# Observation of a spontaneous anomalous Hall response in the Mn$_5$Si$_3$ d-wave altermagnet candidate

Helena Reichlova [1,2] ✉, Rafael Lopes Seeger[3], Rafael González-Hernández[4,5], Ismaila Kounta[6], Richard Schlitz[1], Dominik Kriegner[1,2], Philipp Ritzinger[1,2], Michaela Lammel[7,8], Miina Leiviskä[3], Anna Birk Hellenes[5], Kamil Olejník[2], Vaclav Petřiček[2], Petr Doležal[9], Lukas Horak[9], Eva Schmoranzerova[10], Antonín Badura[10], Sylvain Bertaina[11], Andy Thomas[1,7], Vincent Baltz[3], Lisa Michez[6], Jairo Sinova[5,12], Sebastian T. B. Goennenwein[1,8], Tomáš Jungwirth[2,13] & Libor Šmejkal[2,5] ✉

Phases with spontaneous time-reversal ($\mathcal{T}$) symmetry breaking are sought after for their anomalous physical properties, low-dissipation electronic and spin responses, and information-technology applications. Recently predicted altermagnetic phase features an unconventional and attractive combination of a strong $\mathcal{T}$-symmetry breaking in the electronic structure and a zero or only weak-relativistic magnetization. In this work, we experimentally observe the anomalous Hall effect, a prominent representative of the $\mathcal{T}$-symmetry breaking responses, in the absence of an external magnetic field in epitaxial thin-film Mn$_5$Si$_3$ with a vanishingly small net magnetic moment. By symmetry analysis and first-principles calculations we demonstrate that the unconventional d-wave altermagnetic phase is consistent with the experimental structural and magnetic characterization of the Mn$_5$Si$_3$ epilayers, and that the theoretical anomalous Hall conductivity generated by the phase is sizable, in agreement with experiment. An analogy with unconventional d-wave superconductivity suggests that our identification of a candidate of unconventional d-wave altermagnetism points towards a new chapter of research and applications of magnetic phases.

Anomalous Hall effect (AHE) is a traditional and experimentally convenient tool for identifying phases that spontaneously break $\mathcal{T}$-symmetry[1,2]. The AHE refers to a non-dissipative antisymmetric component of the electrical conductivity tensor, that is odd under $\mathcal{T}$ and that can be generated by certain magnetic orderings[1,3]. Among those, the most common and arguably best understood is the ferromagnetic ordering where the broken symmetries allowing for the AHE are related to the net internal magnetization of the crystal[3]. A common model of ferromagnetism is a collective order in the spin space accompanied by an isotropic partial-wave (s-wave) form of the electronic structure in the momentum space[3,4]. In contrast, anisotropic higher-order partial-wave forms of magnetically ordered phases were elusive and much less is known about their responses[4–7]. In fact, compensated magnetic orderings with a vanishingly small net magnetization have remained outside the scope of the research of the spontaneous $\mathcal{T}$-symmetry breaking responses for more than a century[1,3]. Indeed, these responses can be absent in the conventional compensated antiferromagnets whose spin arrangement on the crystal has a symmetry combining $\mathcal{T}$ with a translation ($\mathbf{t}\mathcal{T}$-symmetry – see Fig. 1a) or with inversion ($\mathcal{PT}$-symmetry)[1].

However, over the past decade, two types of crystal structures were predicted to host the spontaneous $\mathcal{T}$-symmetry breaking responses, including a spontaneous AHE, that are not related to a net internal magnetization of the crystal[8,9]: (i) The first type is geometrically frustrated structures, such as kagome, pyrochlore, or triangular lattices[10–12], where the experimentally observed spontaneous AHE[10–12] was related to a non-collinear magnetic ordering[11] or a spin-liquid state candidate[12]. (ii) For the second type of crystals with a collinear magnetic order, termed altermagnetic[13,14], the distinctive feature are non-relativistic spin symmetries where the opposite-spin sublattices are connected by real-space rotation transformations and not by translation or inversion[1,9,13,14]. In contrast, conventional collinear ferromagnets (ferrimagnets) and antiferromagnets have exclusively distinct symmetries[13,14]: ferromagnets (ferrimagnets) have only one spin lattice (or opposite-spin sublattices not connected by any symmetry transformation), and antiferromagnets have opposite-spin sublattices connected by a real-space translation or inversion. The spontaneous anomalous Hall response in altermagnets has then been related, when including relativistic spin-orbit coupling, to a compensated collinear magnetic order with a vanishingly small (zero non-relativistic) magnetization[1,9,13,14]. The general characteristic of the unconventional magnetism in altermagnets is a strong $\mathcal{T}$-symmetry breaking and alternating spin polarization in both real-space crystal structure and momentum-space electronic structure, with or without the presence of the weak relativistic magnetization[13–17]. The alternating spin polarization has suggested to refer to this phase as altermagnetism[13,14]. Note that, in general, the $\mathcal{T}$-symmetry-breaking responses in altermagnets do not require relativistic spin-orbit coupling[13,14]. In the specific case of the AHE, however, additional symmetry breaking by the spin-orbit coupling is required in collinear magnets, including altermagnets[1,9,14,18]. Experimental confirmation of altermagnetic band structure was recently published[19–22].

Remarkably, for certain crystal symmetries, the altermagnetic phase has been predicted to take a form of an unconventional d-wave magnet[9,14,23]. Unlike the earlier suggested realizations via Fermi-liquid instabilities in strongly-correlated materials[4,6,7], here the d-wave magnetism is generated by a robust crystal potential and an unconventional real-space spin-density ordering[13]. Remarkably, it can also host $\mathcal{T}$-symmetry-breaking responses of comparable strength to the conventional s-wave ferromagnetism[13,14]. Besides the AHE, the predicted responses in these unconventional d-wave magnets also include analogues of the non-relativistic spin-polarized currents that underpin the prominent giant-magnetoresistance and spin-torque phenomena in ferromagnetic spintronic devices[1,9,13,24–29].

In the experimental part of our paper, we present a discovery of a spontaneous anomalous Hall conductivity of 5–20 S/cm in epitaxial thin films of $Mn_5Si_3$ with a vanishingly small net magnetic moment. Our characterization measurements show that the $Mn_5Si_3$ epilayers have a hexagonal crystal structure without canonical geometric frustration. The observed unconventional combination of a spontaneous anomalous Hall response and a vanishingly small net magnetization is, therefore, not related in our $Mn_5Si_3$ epilayers to phases stabilized by the first type of crystal structures with geometric frustration. This turns our attention in the theory section to the second type of crystal structures with the d-wave altermagnetic phase. Below the magnetic-ordering temperature, the crystal structure of $Mn_5Si_3$ is known from previous studies to result in a sizable magnetic moment on two fifths of the Mn atoms, as highlighted in Fig. 1a, b[30,31]. Our first-principles calculations show that without strong correlations, the unconventional d-wave magnetism of these magnetically ordered Mn atoms in the direct real space (Fig. 1b), and the corresponding d-wave spin polarization in the reciprocal momentum space (Fig. 1c), generate a vanishingly small net magnetization and a sizable spontaneous anomalous Hall conductivity of the microscopic Berry-curvature

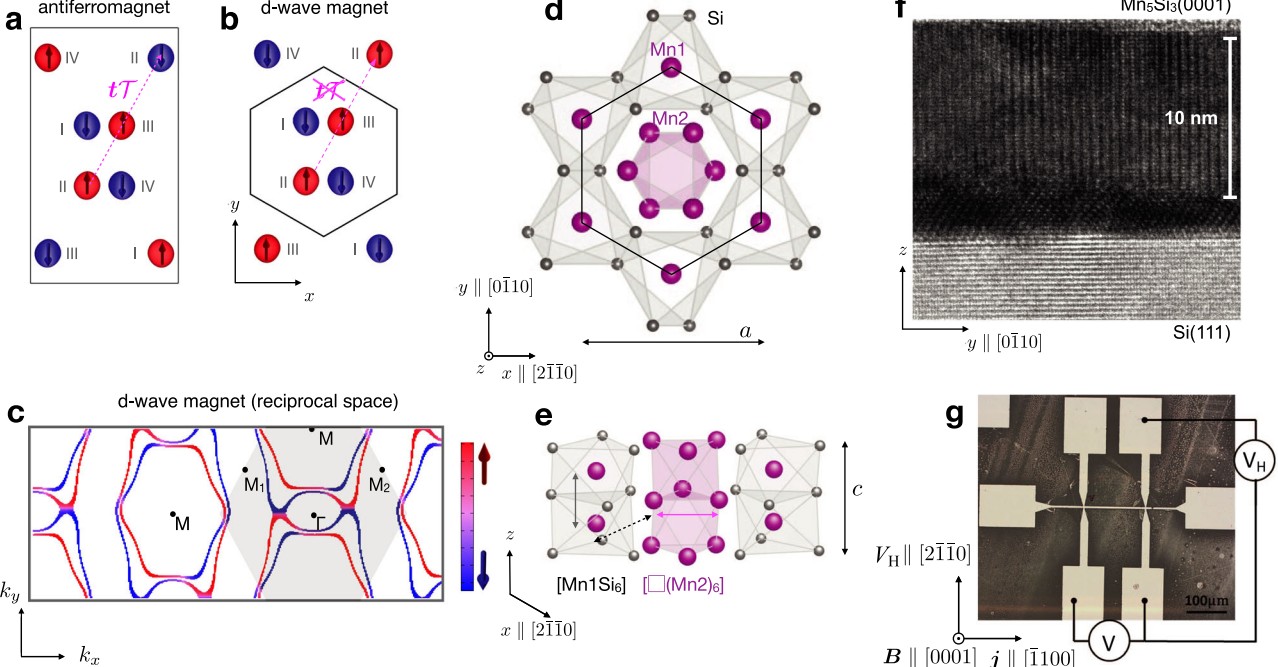

**Fig. 1 | Unconventional d-wave magnetism and crystallographic structure of $Mn_5Si_3$. a** Conventional antiferromagnet with $t\mathcal{T}$ symmetry combining translation with time-reversal. **b** Unconventional d-wave magnetism with broken $t\mathcal{T}$ symmetry. (Opposite magnetization-density isosurfaces calculated from first principles are marked in red and blue.) **c** Spin-split Fermi surface cut of an anisotropic d-wave form calculated from first-principles. The Néel vector is along the [2$\bar{2}$01] crystal direction ([111] direction in the 3-component $a-b-c$ notation), and we plot spin

projection on the [2$\bar{1}\bar{1}$0] $x$-axis ([100] $a$-axis). **d, e** Top and side view, resp., of the hexagonal crystal structure of the $Mn_5Si_3$ epilayers with marked in-plane $a$ and out-of-plane $c$ lattice constants. **f** Transmission electron microscopy image of the $Mn_5Si_3$ epilayer grown on a Si substrate. **g** Optical micrograph of the lithographically patterned Hall bar, and orientation of the crystal and the applied magnetic field **B**.

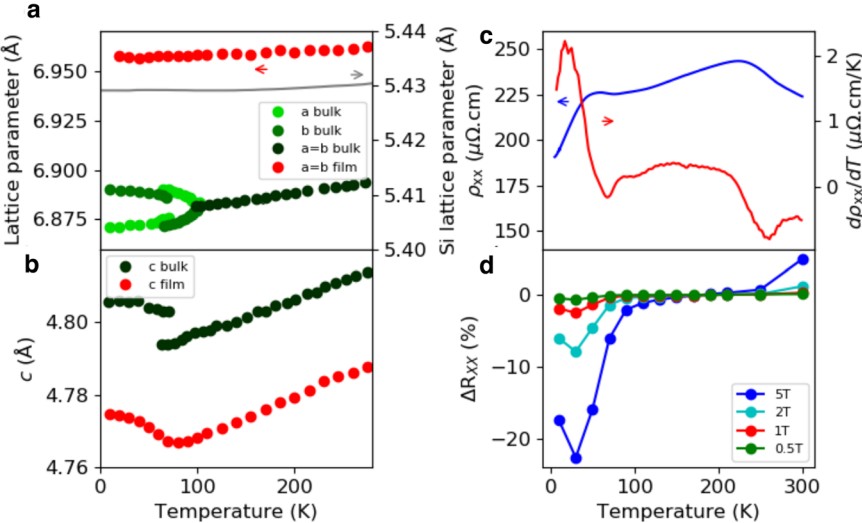

**Fig. 2 | Temperature dependent lattice parameters and resistivity. a, b** In-plane and out-of-plane lattice constants, resp., obtained in our $Mn_5Si_3$ epilayers from temperature dependent X-ray diffraction (red), and compared to previously reported data for bulk $Mn_5Si_3$[30] (green). **c** Temperature-dependent longitudinal resistivity $\rho$ and its derivative $d\rho/dT$. **d** Temperature-dependent longitudinal magnetoresistance recorded with a magnetic field of 0.5,1, 2, and 5 T applied along the [0001] crystal direction.

mechanism[1], consistent with our measurements. We acknowledge that while the proposed magnetic order aligns with our experimental observations, future efforts should be focused on directly observing the spin structure. This is discussed in detail in the Supplementary Note 5. In the final discussion section we point out that our unconventional d-wave magnetism candidate, realized in a crystal comprising abundant and only weakly relativistic elements, points towards research and applications of magnetic band-topology, non-dissipative electronics, valleytronics or spintronics unparalleled within the framework of the conventional ferromagnetic, antiferromagnetic and paramagnetic phases.

## Results

We start the experimental part by discussing the structural characterization of $Mn_5Si_3$ in the room-temperature paramagnetic phase. Earlier studies of bulk crystals determined that the space group of $Mn_5Si_3$ is $P6_3/mcm$, with a hexagonal unit cell containing two formula units[30–32]. The unit cell has sixteen atoms: four Mn atoms (Mn1) at a Wyckoff position 4d, and six Mn atoms (Mn2) and six Si atoms at a Wyckoff position 6g. The crystal structure motif of $Mn_5Si_3$, shown in Fig. 1d, e, is characterized by a distorted octahedron [Mn1Si₆] with Si occupying its vertices and Mn1 in the center, and a distorted octahedron [□(Mn2)₆] with Mn2 at the vertices and no atoms in its interior[30]. Since the distances of Mn atoms in pairs Mn1–Mn1, Mn1–Mn2 and Mn2–Mn2 are substantially different[30], the exchange interactions between Mn atoms do not exhibit the canonical geometric frustration[33].

For our study, we have prepared thin films of $Mn_5Si_3$ by molecular beam epitaxy on top of a Si(111) substrate. In Fig. 1f, we present a room-temperature transmission electron microscopy (TEM) image showing the (0001) orientation of our $Mn_5Si_3$ films with a thickness of 12 nm which were used to fabricate microdevices for electrical transport measurements (Fig. 1g). The TEM measurements, complemented by X-ray diffraction (XRD) shown in the Supplementary Information Figs. 1 and 2 (and also Methods), indicate high crystal quality of the epilayers, with an in-plane hexagonal symmetry. They confirm that our thin films have the same crystal structure motif as previously observed in the bulk samples. Apart from the same crystal-structure motif, there are important differences between the overall crystal structure of the bulk and our thin-film samples that stem from the epitaxial strain and the epitaxial constraints. The $Mn_5Si_3$ epilayers on the Si(111) substrate

are constrained to a hexagonal crystal lattice in the whole studied temperature range and, therefore, the films do not undergo the structural transitions observed in bulk. In the following paragraphs, we elaborate on this point in more detail.

In Fig. 2a, b we show temperature-dependent lattice constants of $Mn_5Si_3$, and we start the discussion by first recalling the behavior of $Mn_5Si_3$ as reported earlier in the bulk samples[30]. The lattice constants $a$ and $b$, that are equal in the room-temperature paramagnetic phase, show two anomalies in bulk $Mn_5Si_3$: one at $T_1 \approx 100$ K and the other one at $T_2 \approx 70$ K (Fig. 2a). At the higher critical temperature $T_1$, the crystal undergoes an orthorhombic distortion that lifts the degeneracy between the $a$ and $b$ lattice parameters. When further decreasing the temperature to the lower critical point $T_2$, a monoclinic distortion results in one of the two lattice parameters abruptly increasing while the other one is decreasing, which is also accompanied by an increase of the lattice parameter $c$ (Fig. 2b).

The structural transitions have counterparts in anomalies at $T_1$ and $T_2$ previously detected in the magnetic susceptibility, specific heat and longitudinal resistivity of the bulk samples[30,31,34]. Earlier neutron scattering measurements on the bulk samples[30,33,35–37] revealed that at $T_1$, the othorhombic crystal distortion is accompanied by an onset of a collinear antiferromagnetic ordering of two-thirds of the Mn2 atoms. The antiferromagnetic propagation vector (0, 1/2, 0) corresponds to a doubling of the unit cell along the $b$-axis, as compared to the paramagnetic phase[30,37]. The resulting $\mathbf{t}\mathcal{T}$-symmetry of this conventional antiferromagnetic phase[13,24] is consistent with the absence of a spontaneous AHE signal[1], as experimentally confirmed in the bulk samples (or thick polycrystalline films)[31,34].

Below $T_2$, the neutron studies in bulk $Mn_5Si_3$ showed that the magnetic phase becomes non-collinear (non-coplanar)[30,36,37]. This second magnetic transition can be suppressed, and the collinear antiferromagnetic phase recovered by an applied magnetic field[30]. The strength of the critical field increases with decreasing temperature, reaching approximately 1 T at 60 K[31,34,36,37]. A spontaneous Hall resistivity of ≈0.02–0.04 $\mu\Omega$ cm measured below $T_2$ in the bulk samples (or thick polycrystalline films) was ascribed[31,34] to a topological Hall effect[38,39] arising from the low-temperature non-collinear magnetic order in $Mn_5Si_3$. Consistently, the topological Hall signal was suppressed by applied magnetic fields of strengths comparable to the above critical fields obtained in the temperature-dependent neutron measurements[30,31,34].

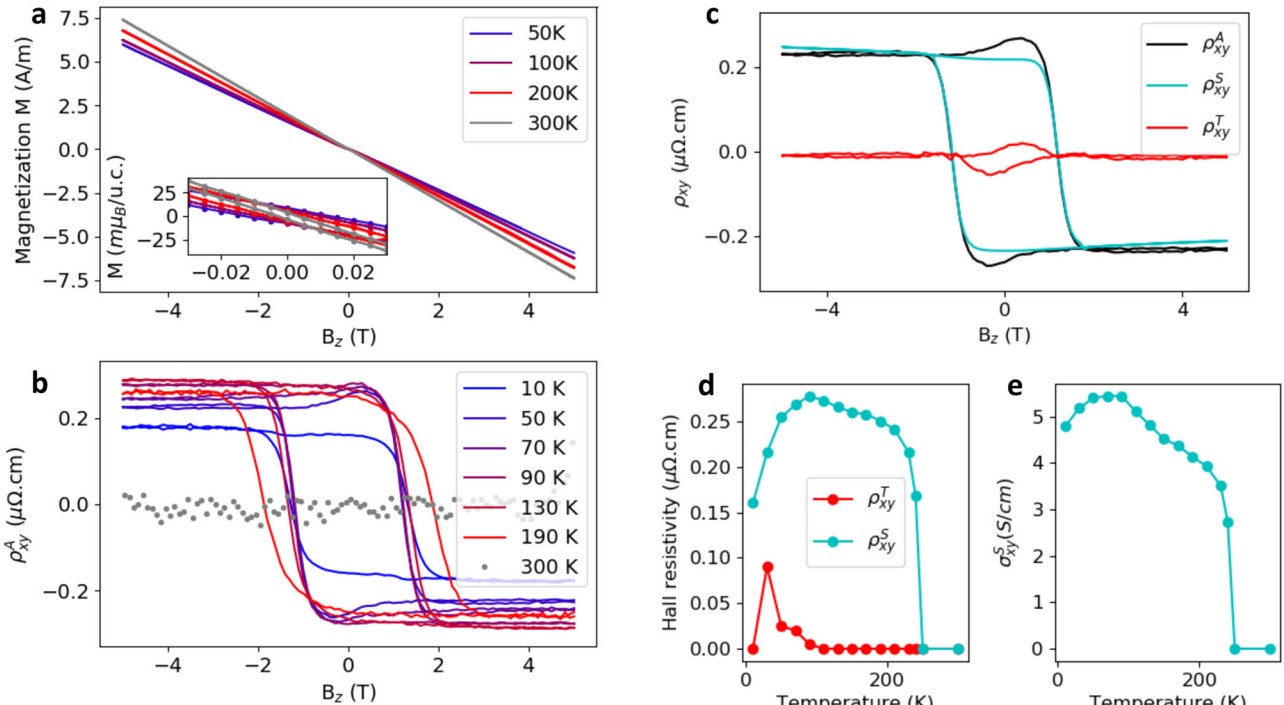

**Fig. 3 | Magnetization and anomalous Hall effect. a** Magnetization as a function of the magnetic field applied along the [0001] crystal direction at different temperatures. The inset highlights a vanishingly small remanent magnetization within error of 5 m$\mu_B$/u.c. The plot shows the total magnetization of the measured sample, i.e., also including the substrate. However, when recalculating from A/m to m$\mu_B$/u.c., we considered that the remanent signal at zero field is due to the $Mn_5Si_3$ film alone and has no contribution from the Si substrate (we also considered the same for the small field range around zero plotted in the inset). **b** Field and temperature dependent anomalous Hall resistivity. The ordinary Hall effect, which is linear in the applied magnetic field, was subtracted (see text). **c** Decomposition of the anomalous Hall resistivity measured at 50 K into a topological Hall component $\rho_H^T$, and a component ascribed to the unconventional d-wave magnetism, $\rho_H^U$ (see Methods and Supplementary Fig. 4).). **d** The decomposition of the spontaneous (zero-field) anomalous Hall resistivity as a function of temperature. **e** Spontaneous anomalous Hall conductivity corresponding to the component ascribed to the unconventional d-wave magnetism.

We now compare the established temperature-dependent phenomenology in bulk $Mn_5Si_3$ to our measurements in the thin-film epilayers. As expected, the in-plane lattice parameters $a$ and $b$ of our epilayers, constrained by the substrate, show no transitions (Fig. 2a), and their weak temperature dependence closely follows the weakly decreasing in-plane lattice parameter with decreasing temperature of the Si substrate. In contrast, the out-of-plane lattice parameter $c$ of the $Mn_5Si_3$ film is not fixed by the substrate, and we observe an anomaly analogous to the $T_2$ transition observed in the bulk samples (Fig. 2b).

Note that in the case of $Mn_5Si_3$ on Si(111), the value of the in-plane lattice constant is governed primarily by the mismatch in the thermal expansion coefficients of the epilayer and the substrate. During cooling after growth, the mismatch in the thermal expansion coefficients, which are around $2.6 \times 10^{-6}$ K$^{-1}$ and $23 \times 10^{-6}$ K$^{-1}$ in Si and $Mn_5Si_3$, respectively, causes an in-plane tensile strain. At room temperature and below we, therefore, find the in-plane lattice constant in our epilayers to be considerably larger than the bulk value. Consistently, the out-of-plane lattice constant in the epilayers is smaller than in bulk $Mn_5Si_3$. In contrast to the thermal-expansion mismatch, the nominal mismatch of 3.7% between room-temperature in-plane lattice constants of the separate Si(111) and $Mn_5Si_3$(0001) crystals plays a more minor role as it is partially accommodated by a thin MnSi interfacial layer between the Si substrate and the $Mn_5Si_3$ epilayer (see Methods for more details).

In Fig. 2c, d we plot resistivity measurements of our microdevices (Fig. 1g) patterned from the thin-film $Mn_5Si_3$ epilayers. They show a metallic resistivity of the order of magnitude that is consistent with earlier studies of thicker films[31,34]. Consistent with the bulk phenomenology, we detect the $T_2$ anomaly in the resistivity of our thin films, as shown in Fig. 2c. We also point out that our observation in Fig. 2d of a

strong magnetoresistance below $T_2$, contrasting with a negligible magnetoresistance over a broad temperature range above $T_2$, is reminiscent of the sensitivity to the magnetic field of the non-collinear component of the magnetic order in the low-temperature phase of the bulk samples.

Unlike the $T_2$ transition, we observe no counterparts of the bulk anomaly at $T_1 \approx 100$ K in either the structural characterization or resistivity measurements of our $Mn_5Si_3$ epilayers. However, as seen in Fig. 2c, d, we detect a second anomaly in the resistivity of the thin films at $\approx$240 K, accompanied by an enhanced magnetoresistance above this temperature.

To explore the phases of our $Mn_5Si_3$ epilayers over the broad temperature range we performed magnetometry and Hall measurements, summarized in Fig. 3. At 300 K, the magnetization is linear in the external magnetic field $B_z$, that we applied along the out-of-plane [0001] crystal axis (Fig. 3a). At lower temperatures, a weak nonlinearity is observed at small fields. In Supplementary Fig. 3 we show control SQUID measurements of a bare Si(111) substrate (with no deposited epilayer), exhibiting a similar weak low-field non-linearity. The important observation in Fig. 3a is that the remanent zero-field magnetization remains below ~0.01 $\mu_B$ per unit cell at all temperatures, as highlighted in the inset of Fig. 3a.

From the measured total Hall resistivity, $\rho_H^{tot}$, we first extracted the component that is linear in $B_z$, and gives an ordinary Hall coefficient, $R_H \approx 1 - 4 \times 10^{-10}$ m$^3$ C$^{-1}$. Assuming a single-band model, it corresponds to a metallic carrier density, $n \sim 10^{22}$ cm$^{-3}$. The remaining anomalous component of the Hall resistivity is given by, $\rho_H^A = \rho_H^{tot} - R_H B_z$ (see Methods and Supplementary Fig. 4). Remarkably, despite the vanishingly small remanent magnetic moment, we observe a sizable spontaneous AHE signal at zero field over a broad range of temperatures.

The AHE exhibits a large coercivity of ≈2–3 T and, overall, it is not diminished by strong fields. Nevertheless, below $T_2$ and within a -1 T field-range, we observe a weak contribution to the anomalous Hall resistivity that is non-monotonic in the field. To highlight this feature, we decompose in Fig. 3e the anomalous Hall resistivity into two contributions: $\rho_H^A = \rho_H^T + \rho_H^U$. The $\rho_H^T$ component appears below $T_2$, has a spontaneous value reaching 0.09 $\mu\Omega$cm, and vanishes at -1 T. Its phenomenology is thus consistent with the topological Hall effect identified in the bulk $Mn_5Si_3$ samples[31,34,40].

In Supplementary Figs. 5, 6, we compare the field-dependence of the AHE with the longitudinal magnetoresistance. A strong negative magnetoresistance is observed below $T_2$, consistent with the presence of the $\rho_H^T$ contribution to the AHE that has been associated with the deviation of the magnetic order from the fully collinear state. Above $T_2$ where the $\rho_H^T$ contribution is absent and the magnetic order is expected to be collinear, we observe the correspondingly weaker magnetoresistance.

Note that the observed large coercive field of ≈2–3 T at which the AHE reverses is consistent with the absence of a strong net magnetic moment, as detected by SQUID, and with the corresponding weak Zeeman coupling in our compensated magnet. The observed increase of the reorientation field (coercivity) with increasing temperature (see also Supplementary Fig. 7) is another signature that contrasts with the conventional ferromagnetic phenomenology. In the collinear compensated magnets, the increasing reorientation field with increasing temperature was already reported in earlier studies and ascribed to a complex and highly anisotropic response to the applied magnetic field[41]. This was associated, besides the magnetic anisotropy and exchange interaction, with the effect of the Zeeman coupling of the field-induced or weak-relativistic net magnetic moment.

The $\rho_H^U$ contribution to the measured anomalous Hall resistivity in our thin-film epilayers is detected below a transition temperature of ≈ 240 K (Fig. 3d), which coincides with the temperature of the second anomaly observed in the resistivity and magnetoresistance measurements in Fig. 2c, d. The $\rho_H^U$ component dominates the anomalous Hall resistivity over the entire temperature range down to the lowest measured temperature of 10 K, i.e., also below $T_2$ (Fig. 3d). The zero-field spontaneous value of $\rho_H^U$ reaches 0.2–0.7 $\mu\Omega$ cm (see also Supplementary Fig. 8 for data measured on other $Mn_5Si_3$ epilayer samples).

In Fig. 3f we plot the spontaneous anomalous Hall conductivity component $\sigma_H^U \approx \rho_H^U/\rho^2(B_z = 0)$ over the whole measured temperature range. The magnitude of $\sigma_H^U$ reaches values between 5 and 20 S/cm, depending on the studied $Mn_5Si_3$ epilayer (see also Supplementary Fig. 8).

To further explore the origin of the $\rho_H^U$ signal, we prepared a series of films with a varying nominal thickness, which gives a varying crystal quality. For film thicknesses ≳ 30 nm, spurious phases are formed in our $Mn_5Si_3$ films. We have, therefore, focused on lower thicknesses and parametrized the quality of the crystals by XRD measurements. In Supplementary Fig. 8 we show that the magnitude of $\rho_H^U$ decays with lowering the crystal quality, which we characterize by the ratio of intensities of $Mn_5Si_3$ and $MnSi$ X-ray diffraction peaks, and the signal is absent in polycrystalline films. We also show that there is no correlation between $\rho_H^U$ and the negligible net magnetic moment measured across the series of samples with varying crystal quality. Finally, we show in Supplementary Fig. 9 measurements in rotating 4 T fields that highlight an unconventional anisotropy of the AHE in our $Mn_5Si_3$ films.

To summarize the experimental part of our work, we observe in our $Mn_5Si_3$ thin-film epilayer a sizable spontaneous AHE signals below ≈240 K, accompanied by a vanishingly small remanent net magnetization. This excludes AHE mechanisms analogous to conventional ferromagnets or due to a field-induced magnetization. The structural characterization of our films implies that exchange interactions between Mn atoms do not exhibit the canonical geometric frustration.

This allows us to further exclude AHE mechanisms associated with compensated magnetically-ordered or spin-liquid phases generated by the geometrically frustrated lattices. We detect a component in our spontaneous Hall response with analogous phenomenology that in bulk $Mn_5Si_3$ samples was ascribed to the topological Hall effect. In contrast to the bulk samples, however, it only represents a weak contribution to the total spontaneous Hall signal measured in our thin-film epilayers. Moreover, the dominant contribution in our samples persists also at temperatures well above $T_2 \approx 70$ K, at which the topological Hall effect disappears in both bulk and our thin films.

The origin of the dominant contribution to the spontaneous AHE in our $Mn_5Si_3$ epilayers is, therefore, unconventional. The following section presents our theory analysis showing that the above structural and electric characterizations, combined with the observation of the spontaneous AHE signal and a vanishingly small net magnetization, are consistent with the formation of the unconventional d-wave magnetic phase (Fig. 1b, c).

## Theory

Earlier density functional theory (DFT) calculations[33] showed that for the two thirds of Mn2 atoms contributing to the magnetic ordering in $Mn_5Si_3$, the strongest exchange coupling, in the notation of Fig. 1a, is between crystal sites I and II, and between sites III and IV. These exchange interactions tend to stabilize the collinear antiparallel ordering[33], consistent with the transition from the paramagnetic to the antiferromagnetic phase observed in the neutron measurements on bulk samples at $T_1 \approx 100$ K[30,37].

In our thin-film epilayers, the spontaneous anomalous Hall signal occurring below ≈ 240 K evidences a transition to a $\mathcal{T}$-symmetry broken phase. On one hand, the close similarity between the crystal-structure motifs identified in our thin films and in the bulk samples suggests that the leading exchange interactions in the thin films are again between the Mn2 sites I and II, and sites III and IV. This implies that a good candidate for the $\mathcal{T}$-symmetry broken phase below ≈ 240 K in the thin films, illustrated in Fig. 1b, has the same antiparallel ordering between the sites I and II, and the sites III and IV, as in the bulk (Fig. 1a). On the other hand, the doubling of the unit cell, and the resulting $\mathbf{t}\mathcal{T}$-symmetry of the conventional antiferromagnetic phase[1,13,24] observed in the bulk samples (Fig. 1a)[30,37], is excluded in our thin films by the experimentally detected spontaneous anomalous Hall signal[1,13,24]. Therefore, the candidate phase of the thin films below ≈240 K (Fig. 1b), consistent with the earlier theoretical and experimental works in bulk samples, and with the complete set of our structural characterization, resistivity and Hall measurements, shares with the bulk samples the antiparallel ordering of the Mn2 magnetic moments while, simultaneously, keeping the same size of the unit cell upon the transition from the paramagnetic to the magnetically ordered phase.

In Fig. 1b we highlight on our DFT real-space magnetization densities, and in Figs. 1c and 4a, b on DFT momentum-space Fermi surfaces, that the candidate magnetic ordering corresponds to the unconventional compensated collinear magnetic phase of the d-wave form[13,24]. In real space, the candidate magnetic ordering shows the defining characteristics of the unconventional phase, dubbed altermagnetic: Namely the lack of translation or inversion and, in the non-relativistic limit, the presence of rotation symmetry transformations connecting opposite-spin sublattices. The rotation symmetries protect the compensated nature of the magnetic phase, i.e. the precisely zero net spontaneous magnetization in the non-relativistic limit, while allowing for the $\mathcal{T}$-symmetry breaking and alternating spin splitting in the band structure[13,24].

Following the general classification of non-relativistic collinear magnetic phases based on the spin-group formalism[13,24], the spin-dependent band structure of our candidate unconventional magnetic phase in $Mn_5Si_3$ is described by a spin Laue group $^2m^2m^1m$. The group is generated by the following three symmetry transformations: a

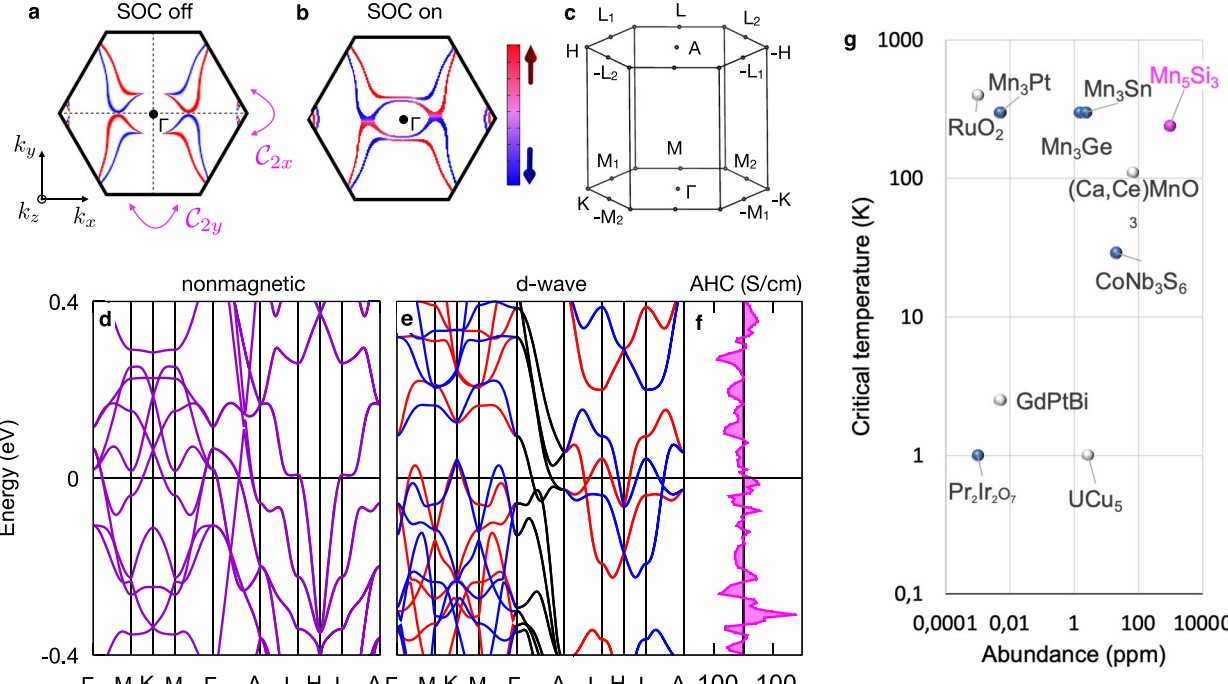

**Fig. 4 | First-principle theory of Mn₅Si₃ electronic structure and comparison chart of compensated magnets with anomalous Hall effect.** Spin-polarized Fermi surface calculated without (**a**) and with (**b**) spin–orbit coupling (SOC). In the latter case, the Néel vector is along the [2̄2̄01] crystal direction ([111] direction in the 3-component *a–b–c* notation), and we plot spin projection on the [21̄1̄0] *x*-axis ([100] *a*-axis).**c**, Hexagonal Brillouin zone notation. Calculated non-relativistic non-magnetic (**d**) and magnetic (**e**) energy bands. The red and blue coloring corresponds to the spin up and down projection, respectively. **f** Calculated anomalous

Hall conductivity $\sigma_H$. **g** Comparison diagram of selected anomalous Hall systems in the parameter space of critical temperature vs. abundance of the least abundant chemical element in the material[59]. Gray colored circles mark systems with the Hall effect mechanism that requires the application of an external field. Blue colored circles mark systems with the spontaneous anomalous Hall effect. The magenta colored circle marks out present study of Mn₅Si₃. Data in panel (**g**) are taken from a review article in ref. 1.

real-space inversion connecting same-spin sublattices, and real-space two-fold rotations around the *x* and *y* axes ($\mathcal{C}_{2x}$ and $\mathcal{C}_{2y}$) connecting opposite-spin sublattices. These spin Laue group symmetries result in two orthogonal spin-degenerate nodal planes crossing the Γ (M) point of the Brillouin zone. When making a closed loop in the momentum space around the Γ (M) point in a plane orthogonal to the spin-degenerate nodal planes, the spin makes a discrete 180° reversal when passing through each nodal plane. This results in the d-wave form of the non-relativistic spin-dependent band structure[13,24], as highlighted on the $k_z = 0$ Fermi-surface cut in Fig. 4a.

By comparing Fig. 4a, b, we see that the relativistic spin-orbit coupling generates only a weak perturbative correction in the Mn₅Si₃ Fermi surfaces. The d-wave form is preserved, and only the discrete 180° spin reversals when passing through the non-relativistic spin-degenerate nodal planes are replaced in the presence of the spin-orbit coupling by a continuous 180° spin reorientation. Note that in the relativistic calculations we considered the magnetic order vector pointing along the [2̄2̄01] crystal direction (for more details see the discussion below on the DFT AHE calculations and Supplementary Note 4).

In Fig. 4c–e we show the Brillouin zone notation and non-relativistic band structure calculations. In the non-magnetic bands, we observe a number of van Hove singularities around the Fermi level. Additonally, in the $k_z = \pi$ plane, the bands are four-fold degenerate. (The origin of the degeneracy is the off-centered mirror plane orthogonal to the *z*-axis[42].) These band structure features make Mn₅Si₃ highly susceptible to the emergence of magnetic ordering[6,13,23].

The stability of the unconventional d-wave magnetic phase can be illustrated based on zero-temperature DFT calculations of the total energy. The d-wave magnetic phase has the total energy smaller than the paramagnetic (ferromagnetic) phase by 0.95 (0.96) eV per unit cell.

Apart from the significantly higher total energies, we also recall that the two reference phases are inconsistent with the set of our experimental results. Namely, the ferromagnetic phase allows for a spontaneous AHE that, however, originates in this phase from its sizable net magnetization. The paramagnetic phase has a zero net magnetization, but also a zero spontaneous AHE.

This brings us to our DFT calculations of the AHE conductivity generated by the compensated magnetic order of the unconventional d-wave phase. In our calculations, we consider the intrinsic microscopic AHE mechanism due to Berry curvature in the relativistic band-structure of a perfect crystal without extrinsic disorder[3,43]. The focus on the intrinsic AHE is justified by the systematic studies of the microscopic mechanisms in ferromagnets[3]. The studies showed that the extrinsic (skew scattering) contribution becomes significant only in samples with conductivities above $10^6$ $\Omega^{-1}$ cm$^{-1}$[3], which is much higher than the conductivity of our Mn₅Si₃ films.

In Fig. 4f, we plot the DFT AHE conductivity as a function of the position of the Fermi level. The calculations are performed for the magnetic order vector pointing along the crystal direction [2̄2̄01] ([111] in the 3-component *a–b–c* notation) between the in-plane [2̄2̄00] and normal-to-the-plane [0001] crystal axes. This off high-symmetry direction is chosen because it gives in our DFT calculations a lower total energy than the in-plane or normal-to-the-plane axes (see Supplementary Note 4). Moreover, the magnetic point group $\bar{1}$ for the Néel vector along the [2̄2̄01] direction allows for a spontaneous anomalous Hall vector component along the [0001] crystal axis, i.e. along the normal to the thin-film plane, which makes it detectable in our experimental geometry. In contrast, AHE is excluded by symmetry in the magnetic point group *mmm* which corresponds in our case to the theoretically identified [0001] hard axis of the N'eel vector. Also consistently with our measurements and DFT calculations, no

spontaneous AHE would be detected for the Néel vector within the (0001)-plane ($c$-plane), ($2\bar{1}\bar{1}0$)-plane ($a$-plane) or ($0\bar{1}10$)-plane ($b$-plane) because in these cases the Hall vector, if allowed, is constrained by symmetry to the (0001)-plane of the thin film.

Our calculations in Fig. 4f illustrate that the spontaneous AHE conductivity, arising from the strong $\mathcal{T}$-symmetry breaking in the electronic structure by the compensated collinear magnetic order of the unconventional d-wave phase, combined with the relativistic Berry-curvature mechanism, can reach values comparable to the AHE in common ferromagnets[3]. We obtain sizable $\sigma_H^U \approx 5 - 20$ Scm$^{-1}$ within a ~100 meV energy window around the Fermi level. These theoretical values are consistent with our measurements.

## Discussion

We have presented our discovery in epitaxial thin-film Mn$_5$Si$_3$ of an unconventional combination of a sizable spontaneous AHE and a vanishingly small net remanent magnetization. Among the experimentally established or theoretically proposed mechanisms, our set of characterization experiments, AHE measurements and DFT calculations is consistent with the formation of the recently predicted unconventional collinear compensated magnetic phase[13,24]. Here we point out that our work complements, in a fundamentally distinct way, parallel experimental AHE studies of other candidate materials of this unconventional phase, namely of RuO$_2$ and MnTe[18,44]. In both RuO$_2$ and MnTe, non-magnetic atoms play a central role in breaking the translation and inversion symmetries, while preserving a rotation symmetry, connecting the opposite-spin sublattices[9,18,44]. Mn$_5$Si$_3$ is principally distinct, as here these basic crystal-symmetry conditions for the formation of the unconventional phase are fulfilled by the spin and crystal arrangement of the magnetic atoms alone. The emergence of the magnetic state we propose has the potential to inspire further research efforts. This is discussed in detail in the Supplementary Note 5.

We also point out that in the experiments on RuO$_2$, the magnetic order vector was reoriented by an applied magnetic field from the zero-field direction to allow for the AHE[18]. In contrast, the measured AHE signal in our thin-film Mn$_5$Si$_3$ is spontaneous, i.e., is observed at zero applied magnetic field. In MnTe, the detected AHE signal was also spontaneous[44] as in the present study. However, the AHE in MnTe was ascribed to a higher-order (g-wave) form of the unconventional magnetic phase[44]. The high-order partial-wave forms can be less favorable as they, e.g., exclude by symmetry giant-magnetoresistive or spin-torque phenomena that are based on non-relativistic spin-dependent conductivities[13,14,24]. In contrast, these prominent $\mathcal{T}$-symmetry-breaking spintronic responses are allowed in the unconventional compensated collinear magnets of the d-wave form, and have been predicted to reach comparable strengths to ferromagnets[13,14,24]. This underlines the foreseen impact in spintronics of our discovery of the spontaneous $\mathcal{T}$-symmetry breaking response in a d-wave magnet candidate[13,24].

The potential implications of the emerging unconventional d-wave magnetism go, however, well beyond the field of spintronics[24]. One area, highlighted by the specific band structure of the d-wave phase of Mn$_5$Si$_3$, is related to the spin splitting of alternating sign at time-reversal invariant momenta (TRIMs). In Mn$_5$Si$_3$, the spin-split TRIMs are the $\pm M_1$ and $\pm M_2$ points in the Brillouin zone, as seen in the DFT band structure calculations in Fig. 4 and confirmed by the spin-group symmetry analysis[24]. The TRIMs in centro-symmetric crystals are known to encode the information on whether the systems can host topological magnetic phases and phenomena[45]. Specifically, they are directly relevant for the axion insulators, topological magneto-electrics, Weyl fermions or the quantum AHE[46]. Moreover, spin polarized valleys centered around the TRIMs of the d-wave magnet represent $\mathcal{T}$-symmetry broken counterparts of the relativistic spin-split valleys in non-magnetic systems, where the $\mathcal{T}$-symmetry excludes spin splitting at TRIMs. The unconventional d-wave phase may thus also open a new research path of unconventional magnetic valleytronics[24].

Next, we look at the d-wave magnet candidate Mn$_5$Si$_3$ from a more practical perspective. In Fig. 4g we show a diagram comparing Mn$_5$Si$_3$ with representative non-collinear and collinear magnets in which the combination of the AHE and the vanishingly small (weak) net magnetization has been experimentally identified. The axes describe the abundance of elements forming the crystals and the magnetic transition temperature. We see that our Mn$_5$Si$_3$ d-wave magnet candidate represents a combination of an exceptional abundance of the involved elements, and a sizable transition temperature.

Finally, we point out that the lighter more abundant elements have a weaker relativistic spin-orbit coupling and tend to make electronic-correlation effects less prominent than heavier elements. Mn$_5$Si$_3$ is thus an example showing that the unconventional d-wave magnetism can be a robust phase not relying on complex strongly relativistic or correlated physics[13,24].

## Methods

### Epitaxial crystal growth

We have grown the epilayers by ultrahigh-vacuum molecular beam epitaxy (MBE) with a base pressure less than $10^{-10}$ Torr. We have cleaned the Si(111) substrate surface by using a modified Shiraki method[47]. We have formed a final oxide layer chemically to protect the Si surface against oxidization in ambient air. This thin oxide layer was then thermally removed by annealing at 900 °C during a few min in the MBE chamber. Subsequently, a 10 nm-thick Si buffer layer was deposited at 600 °C to ensure a high-quality starting surface. The surface of the sample was monitored in situ by the reflection high energy electron diffraction (RHEED) technique that revealed an atomically flat surface with a well-developed (7 × 7) reconstruction (see Supplementary Note 1 and Supplementary Fig. 1). We decreased the growth temperature to 170 °C for the subsequent deposition of Mn and Si. We have evaporated high-purity Mn and Si by using a conventional high-temperature effusion sublimation cells. We have calibrated the cell fluxes by using RHEED oscillations and a quartz microbalance to achieve the desired stoichiometry of the layers with a total growth rate in the range of 0.1–0.2 Å/s. The first monolayers exhibited the typical signature of a Mn$_5$Si$_3$-type crystal, a ($\sqrt{3} \times \sqrt{3}$)R30° reconstruction[48]. Crystal quality was further improved by thermal annealing with its quality degree monitored by RHEED pattern (see Supplementary Fig. 1). Different growth parameters (including the nominal thickness the Mn/Si layers, the Mn and Si deposition rate and the growth temperatures) were optimized to minimize the presence of the spurious MnSi phase. We note that the Curie temperature of MnSi is around 30K and therefore, cannot contribute to the measured signal up to 240K. The same is valid for typical Mn-based oxides which have typically low critical temperature. We show the amount of the spurious phase in our five different samples in Supplementary Fig. 8. We prepared a reference MnSi sample, as discussed in Supplementary Note 3, Supplementary Fig. 10 and we performed reference magnetometry magneto-transport measurements as shown on Supplementary Fig. 11.

### Transmission electron microscopy and X-ray diffraction

TEM investigations were performed at an accelerating voltage of 300 kV on a JEOL JEM-3010 instrument with a spatial resolution of 1.7 Å. The transmission electron microscopy (TEM) cross-section specimens were prepared by using a dual-focused ion beam (FEI Helios 600 NanoLab) milling through a liftout technique. (The TEM analyses summarized in Supplementary Fig. 1 confirmed the epitaxial relationships (Mn$_5$Si$_3$(0001)[10-10]//MnSi(111)[11-2]//Si(111)[1-10]) and reveals the location of MnSi at the interface between the Si substrate and Mn$_5$Si$_3$[49]. The lattice mismatch of 3.7 percent between Si(111) and Mn$_5$Si$_3$ is partially accommodated by the formation of a thin layer of interfacial MnSi and an array of interfacial dislocations. In Supplementary Note 3 we summarize measurements on a control thin

epitaxial film of MnSi deposited on Si(111)[50]. They confirm a negligible role of the MnSi seed layer in our $Mn_5Si_3$/Si(111) films on the measured AHE.

XRD measurements at room temperature were realized using a high brilliancy rotating anode, Rigaku RU-200BH equipped with an image plate detector, Mar345. The radiation used was Cu $K\alpha$, $\lambda$ = 1.5418Å and the beam size was $0.5 \times 0.5$ mm². The high-intensity $Mn_5Si_3$ 0002 reflection in the XRD data recorded at 300K and shown in Supplementary Fig. 2 evidenced the preponderant formation of the $Mn_5Si_3$ hexagonal phase that grows along the c-axis.

Temperature-dependent XRD experiments from which we extracted the lattice constants of our epilayers shown in Fig. 2a, b were performed at CRISTAL beamline of Soleil synchrotron in the Bragg-Brentano geometry using a Siemens D500 diffractometer. The experimental error bar of the data is approximately the size of the dots plotted in Fig. 2a, b. The diffraction-peak intensity in these XRD measurements is much larger compared to the laboratory XRD experiment, as illustrated in Supplementary Fig. 2. Cooling of the sample was provided by a closed-cycle refrigerator (CCR, Sumitomo Heavy Industries), and He exchange gas ensured equalization of the temperature between the cold-finger, thermometer and sample. Cu-$K_{\alpha1,2}$ radiation and a linear detector were used to speed up the data recording[51]. Additional low-temperature XRD measurements have been carried out at 12.7 keV using a 6-circle diffractometer with an angular accuracy better than 0.001°. A 2D XPAD detector and an Advanced Research System closed-cycle cryostat were used in this setup.

## Magnetotransport and anomalous and topological Hall extraction

We have patterned the Hall bars by standard optical lithography and Argon plasma etching. In Supplementary Fig. 5, we show the raw transversal and longitudinal resistivity data, measured simultaneously. In Supplementary Figs. 4 and 6, we show the subtraction of the linear slope, i.e. the ordinary Hall effect. The measured data were separated into symmetric and antisymmetric components. In Fig. 4a we show only the antisymmetric part. This procedure removes the small constant offset in transverse resistivity caused by tiny misalignments of the Hall contacts and the even contribution to the transverse signal. The even contribution presumably originates from anisotropic magnetoresistance due to the low symmetry[52,53]. The anomalous and topological Hall resistivities were extracted by fitting a cosh function. The anomalous Hall contribution is taken as the amplitude of the cosh fit. We show the result in Fig. S3. The additional bump-like features correspond to the topological Hall signal[31]. The recalculated amplitude of the AHE reaches 5-20 S/cm and correlates with the quality of the crystal, as shown in Supplementary Fig. 8. The error of the magnetotransport is caused by thermal noise and current source noise and it is negligible. We provide more details on magneto-transport measurements in Supplementary Note 2.

## Magnetometry measurements

For the magnetic characterization of the $Mn_5Si_3$ thin films, a Quantum Design MPMS7-XL SQUID magnetometer with a reciprocating sample option has been used. The unpatterned sample was cleaned prior to the measurement and mounted using plastic straws. The field-dependent magnetization has been measured at different temperatures for magnetic field strengths between ± 5 T (cp. Fig. 4b) applied out of the sample plane. The signal is dominated by the diamagnetism of the silicon substrate, this diamagnetic contribution is, however, negligible in the small magnetic field (inset). The error of the magnetometry measurements is relatively large because of subtracting the signal from the substrate and the sample holder and we estimate it to be 5 $\mu_B$/u.c.

## Magnetic relativistic density functional theory calculations

The density functional theory calculations were performed using the VASP package[54] employing the projector augmented plane wave method[55]. We have set the energy cut-off of the plane wave basis at 520 eV, used the PBE exchange-correlation functional[56], and the wavevector grid $9 \times 9 \times 12$. For the calculations presented in the main text we have used the in-plane high-temperature lattice constant a=6.902Å[30] and the c-lattice constant corresponding to the bulk collinear phase at T=70 K and our epilayers at T=170 K (4.795Å). Fermi surface calculations of the $Mn_5Si_3$ are shown in Supplementary Fig. 12.

## Berry curvature calculations of Hall conductivity

We have constructed a maximally localized Wannier function and the effective tight-binding model by using the Wannier90 code[57]. We have calculated the intrinsic anomalous Hall conductivity in WannierTools package[58] by employing the Berry curvature formula. We have used a fine-mesh of $320 \times 320 \times 240$ Brillouin zone sampling points and have checked the convergence. Berry curvatures in the $Mn_5Si_3$ are shown in Supplementary Fig. 13.

## Data availability

Data are available from the corresponding authors (H.R. and L.S.) upon reasonable request. We employed the density functional theory code VASP, which can be obtained and purchased at http://www.vasp.at.

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

## Acknowledgements

The authors acknowledge the Czech Science Foundation project no. 22-17899K and Deutsche Forschungsgemeinschaft (DFG) project no. 452301518. Funded by the Deutsche Forschungsgemeinschaft (DFG, German Research Foundation) - 445976410; 490730630 (R.S. and S.T.B.G.). E.S. acknowledges INTER-COST grant no. LTC20026. D.K. acknowledges the Lumina Quaeruntur fellowship LQ100102201 of the Czech Academy of Sciences. H.R. acknowledges Max Planck Dioscuri Program (LV23025). V.P. acknowledges the Czech Science Foundation project no. 18-10504S. K.O. acknowledges the Czech Science Foundation project no. 21-28876J. This work was supported by the French national research agency (ANR) (Project ASTRONICS - Grant Number ANR-15-CE24-0015-01; Project MATHEEIAS, Grant Number ANR-20-CE92-0049-01), and the CNRS International Research Project (IRP) program (Project SPINMAT). T.J. acknowledges Ministry of Education of the Czech Republic Grant No. CZ.02.01.01/00/22008/0004594 and LM202351, ERC Advanced Grant No. 101095925, Czech Science Foundation Grant No. 19-28375X, and the Neuron Endowment Fund Grant.

T.J., J.S. and L.S. acknowledges the EU FET Open RIA Grant No. 766566. A.B.H., J.S. and L.S. acknowledge SPIN+X (DFG SFB TRR 173) and Elasto-Q-Mat (DFG SFB TRR 288). R.G.H. and L.S. gratefully acknowledge the computing time granted on the supercomputer Mogon at Johannes Gutenberg University Mainz (hpc.uni-mainz.de). A.B.H. and L.S. acknowledge Johannes Gutenberg University funding Topdyn, project Alterseed. We acknowledge SOLEIL for provision of synchrotron radiation facilities and we would like to thank Pierre FERTEY for assistance in using the CRISTAL beamline. The low temperature X-ray diffraction was performed in MGML, which was supported within the program of Czech Research Infrastructures (project no. LM2023065).

## Author contributions

H.R. and R.L.S. contributed equally to this work. H.R. and L.S. conceived the idea, H.R., V.B., L.M., S.T.B.G., T.J., and L.S. proposed and supervised the project. H.R. and R.L.S. design the devices and experiments. R.G.H., A.B.H., J.S., T.J., and L.S. analyzed spin and magnetic symmetries and provided first principle calculations. I.K. and L.M. grew the samples, D.K., M.La., V.P., P.D., L.H., E.S., and S.B. characterized the samples, H.R., R.L.S., R.S., P.R., M.Le., K.O., A.B., A.T. did lithography and magneto transport measurements. H.R., T.J., and L.S. wrote the manuscript with contributions from all co-authors.

## Competing interests

The authors declare no competing interests.

## Additional information

[1]Institut für Festkörper- und Materialphysik and Würzburg-Dresden Cluster of Excellence ct.qmat, Technische Universität Dresden, 01062 Dresden, Germany. [2]Institute of Physics, Czech Academy of Sciences, Cukrovarnická 10, 162 00 Praha 6, Czech Republic. [3]Univ. Grenoble Alpes, CNRS, CEA, Grenoble INP, Spintec, F-38000 Grenoble, France. [4]Grupo de Investigación en Física Aplicada, Departamento de Física, Universidad del Norte, Barranquilla, Colombia. [5]Institut für Physik, Johannes Gutenberg Universität Mainz, 55128 Mainz, Germany. [6]Aix Marseille Univ, CNRS, CINAM, AMUTECH, Marseille, France. [7]Leibniz Institute for Solid State and Materials Research (IFW Dresden), Helmholtzstr. 20, 01069 Dresden, Germany. [8]Universität Konstanz, Fachbereich Physik, 78457 Konstanz, Germany. [9]Department of Condensed Matter Physics, Faculty of Mathematics and Physics, Charles University, Ke Karlovu 5, 121 16 Prague 2, Czech Republic. [10]Department of Chemical Physics and Optics, Faculty of Mathematics and Physics, Charles University, Ke Karlovu 5, 121 16 Prague 2, Czech Republic. [11]Aix Marseille Univ, CNRS IM2NP-UMR, 7334 Marseille, France. [12]Department of Physics, Texas A&M University, College Station, Texas, USA. [13]School of Physics and Astronomy, University of Nottingham, NG7 2RD Nottingham, UK. ✉e-mail: reichlh@fzu.cz; smejkall@fzu.cz

