## [Peer Review File · Nature Communications]

REVIEWER COMMENTS

Reviewer #1 (Remarks to the Author):

The authors report the AHE in the collinear AFM phase of d-wave magnet Mn₅Si₃. This AHE in AFM Mn₅Si₃ is indicated to be different from non-collinear AFM Mn₃X and shares the analogous origin with RuO₂ (Nat. Electron. 5, 735 (2022)). Specifically, it is attributed to the Berry phase generated from the spin splitting band structure of Mn₅Si₃, whose antiferromagnetic sublattices are connected by crystal rotation symmetry without the tT symmetry. The results are interesting and may promote antiferromagnetic materials in spintronics, valleytronics, and practical use as the authors claimed. However, the authors spent a lot of time to explain the inconsistency between their results on thin-film Mn₅Si₃ and former bulk Mn₅Si₃, but seems not convincing enough. The reviewer also has some comments on the MnSi spurious phase, M-H, R-H measurements, and DFT calculations. The authors should carefully handle them before further consideration.

1. The authors carried out substantial lattice constant measurements to explain the differences between thin film and bulk Mn₅Si₃, which is consistent with the breaking of tT symmetry and the resultant observation of AHE in the collinear AFM phase of thin film Mn₅Si₃. However, the reasons for these differences are missing. Does epitaxial strain contribute to these differences? Indeed, as shown in Fig. 2b, the difference of c constant between bulk and thin film are quite large. Does this difference consistent with the strain in epitaxial Mn₅Si₃ on Si(111)? Or the magnetic structure of thin-film Mn₅Si₃ is totally different from that of bulk Mn₅Si₃? Besides, the X-Ray diffraction intensity of Mn₅Si₃(0002) is on the order of 100 as shown in Fig. S2, which brings about the question about the reliability of the lattice constant.

2. As shown in Fig. S1, MnSi is used as a seed layer to grow Mn₅Si₃, which is a typical ferromagnet. Is the measured AHE rooted from MnSi? The authors may argue that the Curie temperature of MnSi is only ~30 K. However, in analogous to the increase of T₁ of Mn₅Si₃ to above 200 K, the Curie temperature of MnSi may also increase due to thin-film strain. At least, the measured topological Hall effect in Fig. 3c may be contaminated by MnSi. Besides, to extract the ρ_{XX} for Mn₅Si₃, the ρ_{XX} for MnSi should also be measured and presented. The authors should carry out additional control experiments to clearly clarify these points.

3. For the M-H measurement, the authors claim that “a weak non-linearity is observed at small field”. What is the origin of this weak magnetism in the collinear AFM phase of Mn₅Si₃? Does it contribute to the AHE signal? Besides, the measurement temperature of M-H and R-H shown in Fig. 3a and Fig. 3b should be the same for comparison.

4. For the R-H measurement, why the coercivity occurs at 2-3 T? Does a spin-flop transition occur at corresponding magnetic field? Moreover, why is the coercivity of R-H at 190 K larger than that of 90 K? Based on my simple comprehension, the magnetic order should be weaker at higher temperature, which means the magnetocrystalline anisotropy should be lower, consistent with a lower coercivity. In Fig. S5, sample 2 has the highest ρ_{xy} with middle ρ_{yx} , which means ρ_{xx} varies strongly between samples. What is the origin of this variation? Does the roughness of films vary much with different MnSi components to contribute this variation?

5. For the DFT calculations, the authors claim “The calculations are performed...in plane and normal-to-the-plane” in Page 15. What is the exact direction of “in plane”? Moreover, for deriving AHC through DFT calculation, the band structure with SOC is necessary. The authors should present these data to compare them with the ones without SOC and explain their possible differences.

Reviewer #2 (Remarks to the Author):

In this manuscript the authors present a combination of thorough and extremely interesting transport and magnetization measurements (with supporting structural characterization) on epitaxial Mn₅Si₃ films, and theoretical calculations on the same material. The presentation is very clear and most details needed are presented. The theory predictions and experimental evidence, which match nicely, show evidence of a unique state with effectively zero magnetization, but a sizable anomalous Hall effect, which the authors interpret as evidence of a d-wave magnetic state. This topic of altermagnetism is very new and making a splash, and to my eye this paper makes an important contribution to this field and I feel the case for publication in Nature Communications is strong. It is especially interesting that the relevant phase transition temperature seems quite close to room temperature, making potential real-world applications more credible than for some novel materials. I do have a few comments and questions, but these mostly focus on issues of presentation or minor clarifications, and can be considered by the authors if they feel they would strengthen an already very impressive paper.

Comments/questions:

1. The authors might consider clarifying in the caption of Fig. 1 that panels a and b represent real space, while panel c represents reciprocal space. One is tempted to start matching the patterns of red and blue in these figures, which is not the right thing to do (and is confusing since the pattern in

1c does not match 1b). This is clarified in the main text at the beginning of the theory section, but that is found only after reading for some time...

2. At the top of page 9 the authors point out that the thin epitaxial films have a similar electrical resistivity to a batch of thicker polycrystalline films. This statement may need a bit of clarification or context. It seems possible to me that this congruence is more coincidence than anything. The epitaxial film should seemingly have less scattering from grain boundaries compared to a polycrystal, but this is perhaps then offset by the low thickness? If the authors think there is something important about these similar (and quite high) resistivities, they may need to clarify.

3. In the second complete paragraph on page 10, the authors state that " that the magnitude of ρ_{Uxy} decays with lowering the crystal quality." This is a somewhat imprecise statement, could the authors be more clear about what exactly they mean by "lower crystal quality" here? It also seems that the experimental parameter they changed to vary crystalline quality (which I think is the ratio between the desired phase and an impurity phase?) is film thickness, but these thicknesses are not ever given so far as I can tell? Can the authors add this information?

4. In the caption of figure 4, the authors state that the data point communicating the results for the material discussed in this manuscript is turquoise, but I believe in the figure it is magenta?

5. The temperature dependence of the longitudinal magnetoresistance is part of the evidence presented for the d-wave state. However, it might be interesting to see the magnetic field dependence of this longitudinal magnetoresistance at a fixed temperature, to compare to the AHE data. Do the authors have this data?

Reviewer #3 (Remarks to the Author):

In this work, the authors have reported that an anomalous Hall effect (AHE) emerges in the thin film of Mn_5Si_3 , which is attributed to an unconventional "d-wave" magnetism of this film. After reading the manuscript carefully, it turns out that the concept "d-wave" is used to denote the collinear antiferromagnets with nonrelativistic spin splitting in the band structure. The experimental measurements of the AHE are comprehensive.

So far, the AHE in antiferromagnets are well understood and many antiferromagnetic systems hosting AHE have been observed. The current work does not show significant advances compared to previous reports, especially in the absence of the solid evidence of the magnetic structure. The authors have spent a lot of efforts to describe the nonrelativistic electronic structures of the “d-wave” magnetism. However, it’s well known that the AHE is a relativistic effect. In this sense, it’s unnecessary to link AHE with a misleading nonrelativistic concept. Therefore, I feel the current work does not meet the high publication standard of Nature Communications.

I have several comments about this work, as shown below:

1. The authors kept emphasizing that Mn₅Si₃ is perfectly compensated with precisely zero net magnetization. This is misleading. Although the strength of AHE and the magnitude of the net magnetization are not coupled, the AHE and the net magnetization have the same symmetry constraints, i.e. an AHE and a net magnetization (though it might be vanishingly small) must exist simultaneously. Therefore, a fully compensated antiferromagnets cannot host AHE. An antiferromagnet with AHE must be uncompensated, and thus can be seen as a “weak ferromagnet”. In Fig.3a, it can be seen that the remanent net magnetization has the similar magnitude as that of Mn₃Sn (Ref. 18). Without this net magnetization, one cannot use a magnetic field to reverse the moments and hence the AHE, as shown in Fig.3 b,c. I suggest the authors revise their manuscript to avoid these misleading arguments. They also need to discuss the direction of the net magnetization, and the mechanism to generate the magnetization (Some canting in the presence of SOC? Some symmetry operation is breaking?), since the Hall pseudovector has the same symmetry constraint as the net magnetization.

2. In the main text, the authors confirmed that the crystal structure of the thin film is the same as that in bulk. And in the beginning of the theory part, they showed that the bulk is an antiferromagnet with the magnetic structure shown in Fig. 1a, and the thin film is “not” antiferromagnetic but a “d-wave magnet” with the magnetic structure shown in Fig. 1b, where the outer Mn-moments are reversed and the inner Mn-moments remain the same compared to these in Fig. 1a. The authors have not shown why the Fig. 1a in bulk changes to Fig. 1b in the thin film, the latter seems to be artificially proposed without any experimental evidence. Theoretically, the authors only compare the calculated energies of Fig. 1b and a ferromagnetic and a nonmagnetic phase, but have not shown the energy of Fig. 1a. Therefore, it’s not clear whether the Fig. 1b is stable compared to Fig. 1a. Moreover, why can the high temperature magnetic order above 100 K be Fig. 1a with some small canting? Or a noncollinear antiferromagnetic order?

3. Is there any special reason to plot the Brillouin zones with different centers? This makes the Fermi surface shown in Fig. 1 and Fig. 3 look different and results in confusions.

4. The authors described the nonrelativistic spin splitting of the band structure in Mn₅Si₃. However, it is not clear how the nonrelativistic spin splitting influences the Berry curvature and hence the AHE.

5. The magnetic order in Fig. 1b supports a c_{2x} rotation, which forbids AHE. This is contradictory to their results.

6. The RuO₂ is also considered to be a “d-wave” by some of the authors in their recent works. In this sense, why are the AHE of RuO₂ and Mn₅Si₃ different, since the AHE is closely linked to the “d-wave” magnetism by the authors?

7. The authors showed the calculation of AHC is performed with a [111] oriented Neel vector. Can this anisotropy be confirmed by experiment?

8. What is the direction of the Neel vector used when calculating the Fermi surface with SOC? What is the spin component shown in the Fermi surface with SOC? If the Neel vector is the low symmetric [111] direction, why are the Fermi surface and the spin distributions so symmetric/antisymmetric?

9. Usually, the magnetism of a material is determined by the magnitudes and alignment of the magnetic moments. In this sense, the Mn₅Si₃ should be an antiferromagnet. In this work, however, the authors proposed the concept “d-wave magnetism” according to the nonrelativistic electronic structure. I think such a concept is unnecessary and may lead to many confusions. For example, everyone in the community of magnetic materials and magnetism will say both Fig. 1a and Fig. 1b are antiferromagnetic, but the current work seems to suggest Fig. 1b is not antiferromagnetic.

In summary, in this work, the authors have reported an AHE in Mn₅Si₃. They claimed that this phenomenon is attributed to the artificially proposed magnetic order in Fig. 1b, which does not have experimental and theoretical evidence. The authors focused on discussing the nonrelativistic electronic structures induced by the magnetic order in Fig. 1b and proposed a new concept “d-wave magnetism”. However, it’s not clear why this nonrelativistic concept is important for the AHE, since AHE is a relativistic phenomenon. I suggest the authors perform additional measurements as possible for the magnetic structure and magnetic anisotropy, and try to explain their observations using conventional languages such as magnetic space/point groups, the symmetry allowed net magnetization and the conductivity tensors, the Berry curvatures, etc.

Reviewer #1

Comment #1

1. The authors carried out substantial lattice constant measurements to explain the differences between thin film and bulk Mn_5Si_3 , which is consistent with the breaking of tT symmetry and the resultant observation of AHE in the collinear AFM phase of thin film Mn_5Si_3 . However, the reasons for these differences are missing. Does epitaxial strain contribute to these differences? Indeed, as shown in Fig. 2b, the difference of c constant between bulk and thin film are quite large. Does this difference consistent with the strain in epitaxial Mn_5Si_3 on Si(111)? Or the magnetic structure of thin-film Mn_5Si_3 is totally different from that of bulk Mn_5Si_3 ? Besides, the X-Ray diffraction intensity of $Mn_5Si_3(0002)$ is on the order of 100 as shown in Fig. S2, which brings about the question about the reliability of the lattice constant.

Response

The epitaxial strain is of the key importance here. To emphasize the role of the strain more explicitly, we have revised the corresponding paragraphs of the main text as follows:

“...They confirm that our thin films have the same crystal structure motif as previously observed in the bulk samples. The difference between bulk and our thin-film samples comes from the epitaxial strain and the epitaxial constraints. Mn_5Si_3 epilayers on the Si(111) substrate are constrained to a hexagonal crystal lattice in the whole studied temperature range and, therefore, the films do not undergo the structural transitions observed in bulk. In the following paragraphs, we elaborate on this point in more detail...”

“We now compare the established temperature-dependent phenomenology in bulk Mn_5Si_3 to our measurements in the thin-film epilayers. As expected, the in-plane lattice parameters a and b of our epilayers, constrained by the substrate, show no transitions (Fig. 2a), and their weak temperature dependence closely follows the weakly decreasing in-plane lattice parameter with decreasing temperature of the Si substrate. In contrast, the out-of-plane lattice parameter c of the Mn_5Si_3 film is not fixed by the substrate, and we observe an anomaly analogous to the T_2 transition observed in the bulk samples (Fig. 2b).

Note that in the case of Mn_5Si_3 on Si(111), the value of the in-plane lattice constant is governed primarily by the mismatch in the thermal expansion coefficients of the epilayer and the substrate. During cooling after growth, the mismatch in the thermal expansion coefficients, which are around $2.6 \times 10^{-6} \text{ K}^{-1}$ and $23 \times 10^{-6} \text{ K}^{-1}$ in Si and Mn_5Si_3 , respectively, causes an in-plane tensile strain. At room temperature and below we, therefore, find the in-plane lattice constant in our epilayers to be considerably larger than the bulk value. Consistently, the out-of-plane lattice constant in the epilayers is smaller than in bulk Mn_5Si_3 . In contrast to the thermal-expansion mismatch, the nominal mismatch of 3.7% between

room-temperature in-plane lattice constants of the individual Si(111) and Mn₅Si₃(0001) crystals plays a more minor role as it is partially accommodated by a thin MnSi interfacial layer between the Si substrate and the Mn₅Si₃ epilayer (see Methods for more details).”

Regarding the X-Ray diffraction intensity of the Mn₅Si₃ (0002) peak we have amended the Methods section by the following explanation:

“Temperature-dependent XRD experiments from which we extracted the lattice constants of our epilayers shown in Fig. 2a,b were performed at CRISTAL beamline of Soleil synchrotron in the Bragg-Brentano geometry using a Siemens D500 diffractometer. The experimental error bar of the data is approximately the size of the dots plotted in Fig. 2a,b. The diffraction-peak intensity in these XRD measurements is much larger compared to the laboratory XRD experiment, as illustrated in Supplementary Fig. S2 ...”

Comment #2

2. As shown in Fig. S1, MnSi is used as a seed layer to grow Mn₅Si₃, which is a typical ferromagnet. Is the measured AHE rooted from MnSi? The authors may argue that the Curie temperature of MnSi is only ~30 K. However, in analogous to the increase of T₁ of Mn₅Si₃ to above 200 K, the Curie temperature of MnSi may also increase due to thin-film strain. At least, the measured topological Hall effect in Fig. 3c may be contaminated by MnSi. Besides, to extract the pXX for Mn₅Si₃, the pXX for MnSi should also be measured and presented. The authors should carry out additional control experiments to clearly clarify these points.

Response

To address this comment we have grown and characterized a control thin (8 nm) epitaxial film of MnSi on Si(111). We observe a ferromagnetic transition temperature below 50 K, consistent with earlier studies. We detect an anomalous Hall effect in this MnSi thin film below 50 K in an applied out-of-plane field. It shows the characteristic non-hysteretic hard-axis field-sweep dependence. This confirms that the thin MnSi seed layer in our epitaxial Mn₅Si₃ films cannot explain the dominant characteristics of the anomalous Hall signal, including the remanence and the ≈ 2-3 T coercivity, observed over the broad temperature range.

From the measured resistivity of the control MnSi thin film we conclude that less than 10 % of the current in our Si(111)/Mn₅Si₃ films is shunted by the MnSi seed layer.

The results based on the control Si(111)/MnSi thin film are summarized in the revised Supplementary Sec. III. In the main text Methods, we have included the following remark:

“In Supplementary Sec. III we summarize measurements on a control thin epitaxial film of MnSi deposited on Si(111). They confirm a negligible role of the MnSi seed layer in our Mn₅Si₃/Si(111) films on the measured AHE.”

Comment #3

3. For the M-H measurement, the authors claim that “a weak non-linearity is observed at small field”. What is the origin of this weak magnetism in the collinear AFM phase of Mn₅Si₃? Does it contribute to the AHE signal? Besides, the measurement temperature of M-H and R-H shown in Fig. 3a and Fig. 3b should be the same for comparison.

Response

Based on control SQUID measurement of a bare substrate, we ascribe the weak non-linearity, whose field and temperature dependences are uncorrelated with the detected anomalous Hall signal, to the substrate. To address this point in the revised main text, we have extended the discussion of Fig. 3a as follows:

“...At lower temperatures, a weak non-linearity is observed at small fields. In Supplementary Fig. S3 we show control SQUID measurements of a bare Si(111) substrate (with no deposited epilayer), exhibiting a similar weak low-field non-linearity. The important observation in Fig. 3a is that the remanent zero-field magnetization remains below $\sim 0.01 \mu_B$ per unit cell at all temperatures, as highlighted in the inset of Fig. 3a.”

The control SQUID measurements of the bare substrate are presented in the revised Supplementary Sec. I.

Following the Reviewer’s comment, Fig. 3a and 3b have been replotted to show the measured M-H and R-H curves for corresponding temperatures.

Comment #4

4. For the R-H measurement, why the coercivity occurs at 2-3 T? Does a spin-flop transition occur at corresponding magnetic field? Moreover, why is the coercivity of R-H at 190 K larger than that of 90 K? Based on my simple comprehension, the magnetic order should be weaker at higher temperature, which means the magnetocrystalline anisotropy should be lower, consistent with a lower coercivity. In Fig. S5, sample 2 has the highest pxy with middle oxy, which means pxx varies strongly between samples. What is the origin of this variation? Does the roughness of films vary much with different MnSi components to contribute this variation?

Response

We have addressed the comment on the coercivity as follows in the revised main text and supplementary:

“Note that the observed large coercive field of $\approx 2-3$ T at which the AHE reverses is consistent with the absence of a strong net magnetic moment, as detected by SQUID, and with the corresponding weak Zeeman coupling in our compensated magnet. The observed increase of the reorientation field (coercivity) with increasing temperature (see also Supplementary Fig. S6 and S7) is another signature that contrasts with the conventional ferromagnetic phenomenology. In the collinear compensated magnets, the increasing reorientation field with increasing temperature was already reported in earlier studies and ascribed to a complex and highly anisotropic response to the applied magnetic field [35]. This was associated, besides the magnetic anisotropy and exchange interaction, with the effect of the Zeeman coupling of the field-induced or weak-relativistic net magnetic moment.”

Regarding the resistivity variation, we have added the following explanation in the revised caption of Supplementary Fig. S8:

“We attribute the variations in the longitudinal resistivity of the studied samples to variations in the contribution to scattering from interface/surface roughness.”

Comment #5

5. For the DFT calculations, the authors claim “The calculations are performed...in plane and normal-to-the-plane” in Page 15. What is the exact direction of “in plane”? Moreover, for deriving AHC through DFT calculation, the band structure with SOC is necessary. The authors should present these data to compare them with the ones without SOC and explain their possible differences.

Response

To clarify the considered direction of the magnetic order vector, we have revised the text as follows:

“The calculations are performed for the magnetic order vector pointing along the crystal direction [2-201] ([111] in the in the 3-component $a - b - c$ notation) between the in-plane [2-200] and normal-to-the-plane [0001] crystal axes.”

Regarding the DFT band-structure calculations, we have revised the text as follows:

“By comparing Fig. 4a and Fig. 4b, we see that the relativistic spin-orbit coupling generates only a weak perturbative correction in the Mn_5Si_3 Fermi surfaces. The d-wave form is preserved, and only the discrete 180° spin reversals when passing through the non-relativistic spin-degenerate nodal planes are replaced in the presence of the spin-orbit

coupling by a continuous 180° spin reorientation. Note that in the relativistic calculations we considered the magnetic order vector pointing along the [2-201] crystal direction (for more details see the discussion below on the DFT AHE calculations and Supplementary information Sec. IV.)”

Reviewer #2

Comment #1

1. The authors might consider clarifying in the caption of Fig. 1 that panels a and b represent real space, while panel c represents reciprocal space. One is tempted to start matching the patterns of red and blue in these figures, which is not the right thing to do (and is confusing since the pattern in 1c does not match 1b). This is clarified in the main text at the beginning of the theory section, but that is found only after reading for some time...

Response

Following this comment, we have added in the revised Fig. 1c an explicit note that the Fermi surface illustrates d-wave magnetism in the reciprocal space. In the main text, we have revised the discussion of Fig. 1b,c as follows:

“Our first-principles calculations show that without strong correlations, the unconventional d-wave magnetism of these magnetically ordered Mn atoms in the direct real space (Fig. 1b), and the corresponding d-wave spin polarization in the reciprocal momentum space (Fig. 1c), generate a vanishingly small net magnetization and a sizable spontaneous anomalous Hall conductivity of the microscopic Berry-curvature mechanism [1], consistent with our measurements.”

Comment #2

2. At the top of page 9 the authors point out that the thin epitaxial films have a similar electrical resistivity to a batch of thicker polycrystalline films. This statement may need a bit of clarification or context. It seems possible to me that this congruence is more coincidence than anything. The epitaxial film should seemingly have less scattering from grain boundaries compared to a polycrystal, but this is perhaps then offset by the low thickness? If the authors think there is something important about these similar (and quite high) resistivities, they may need to clarify.

Response

In the earlier studies of thicker polycrystalline films, the reported resistivities of a 160 nm film were lower than in the 45 nm film. This can be explained by the contribution from the interface scattering that becomes more dominant in thinner films. Our 12 nm epilayer has a

lower resistivity than the 45 nm polycrystalline film which can be due to a combination of a lower interface roughness and a lower density of defects in our epitaxial film. The resistivities in all these films are, nevertheless, of the same order of magnitude. The only point that we highlight here is that our measured resistivities are metallic and consistent with the earlier reports. We do not elaborate on the origin of the relatively small quantitative differences among the films.

To clarify this, we have revised the corresponding paragraph as follows:

“In Figs. 2c,d we plot resistivity measurements of our microdevices (Fig. 1g) patterned from the thin-film Mn_5Si_3 epilayers. They show a metallic resistivity of the order of magnitude that is consistent with earlier studies of thicker films [26,28]....”

Comment #3

3. In the second complete paragraph on page 10, the authors state that " that the magnitude of ρ_{Uxy} decays with lowering the crystal quality." This is a somewhat imprecise statement, could the authors be more clear about what exactly they mean by "lower crystal quality" here? It also seems that the experimental parameter they changed to vary crystalline quality (which I think is the ratio between the desired phase and an impurity phase?) is film thickness, but these thicknesses are not ever given so far as I can tell? Can the authors add this information?

Response

To address this comment, we have added the film thicknesses in the revised caption of Supplementary Fig. S8, and we have modified the sentence in the main text as follows:

“In Supplementary Fig. S8 we show that the magnitude of ρ_{UH} decays with lowering the crystal quality, which we characterize by the ratio of intensities of Mn_5Si_3 and MnSi X-ray diffraction peaks, and the signal is absent in polycrystalline films.”

Comment #4

4. In the caption of figure 4, the authors state that the data point communicating the results for the material discussed in this manuscript is turquoise, but I believe in the figure it is magenta?

Response

We have fixed the typo.

Comment #5

5. The temperature dependence of the longitudinal magnetoresistance is part of the evidence presented for the d-wave state. However, it might be interesting to see the magnetic field dependence of this longitudinal magnetoresistance at a fixed temperature, to compare to the AHE data. Do the authors have this data?

Response

To address this comment, we have added the following paragraph in the main-text discussion of Fig. 3 and the corresponding new Supplementary Fig. S5 and S6 containing the measured magnetoresistances:

“In Supplementary Fig.~S5 and S6, we compare the field-dependence of the AHE with the longitudinal magnetoresistance. A strong negative magnetoresistance is observed below T_2 , consistent with the presence of the ρ^T_H contribution to the AHE that has been associated with the deviation of the magnetic order from the fully collinear state. Above T_2 where the ρ^T_H contribution is absent and the magnetic order is expected to be collinear, we observe the correspondingly weaker magnetoresistance.”

Reviewer #3

General Comment

So far, the AHE in antiferromagnets are well understood and many antiferromagnetic systems hosting AHE have been observed. The current work does not show significant advances compared to previous reports, especially in the absence of the solid evidence of the magnetic structure. The authors have spent a lot of efforts to describe the nonrelativistic electronic structures of the “d-wave” magnetism. However, it’s well known that the AHE is a relativistic effect. In this sense, it’s unnecessary to link AHE with a misleading nonrelativistic concept. Therefore, I feel the current work does not meet the high publication standard of Nature Communications.

Response

Our work is principally distinct from previous studies of compensated magnets whose magnetic structure in bulk is well established (primarily from neutron scattering) and is not considered to significantly change in thin films. In these earlier AHE studies on bulk crystals or thin films, the central focus was on the observation of the AHE and on understanding its seemingly contradicting coexistence with the compensated nature of the known magnetic order in the studied materials.

In contrast, the established magnetic structure of bulk crystals of the compensated magnet Mn_5Si_3 in its collinear antiferromagnetic phase has the translation symmetry transformation connecting the opposite-spin sublattices. This excludes the AHE by symmetry, consistent with earlier Hall measurements in the collinear antiferromagnetic phase of the bulk Mn_5Si_3

crystals. In our work we focus on the identification of a candidate magnetic phase of our compensated thin (12 nm) films of Mn_5Si_3 that, remarkably, show a sizable spontaneous AHE. Since identifying the magnetic order by direct methods like neutron scattering is generally notoriously difficult in films of such a small thickness, we base our work on a set of complementary structural, magnetic and transport measurements, and a comprehensive theoretical analysis.

Because the starting point of our combined experimental and theoretical study is an unknown type of magnetic order, the first and primary focus is on the identification of the candidate magnetic phase whose electronic structure has broken time-reversal (T) symmetry and is consistent with the full set of experimental and theoretical observations. This primary focus stems from the fact that T-symmetry breaking in the electronic structure is a generally necessary condition for the presence of the AHE. Relativistic spin-orbit coupling and Berry curvature play a significant role in our work but, hierarchically, come after the identification of the candidate magnetic phase with T-symmetry broken electronic structure. This is because the relativistic spin-orbit coupling is not generally a necessary condition for the presence of the AHE, and the Berry curvature is generally not the only possible microscopic mechanism. (These general principles are summarized, e.g., in the review article in Ref. [1] of our manuscript).

Within the primary focus of our manuscript on the identification of the candidate magnetic phase with T-symmetry broken electronic structure, let us briefly highlight the following two points that are elaborated on in detail in the manuscript:

(i) The measured crystal structure of our thin-film Mn_5Si_3 confirms that the exchange interactions between Mn atoms do not exhibit the canonical geometric frustration that could lead to a non-collinear magnetic ordering. This is consistent with the reported collinear antiferromagnetic ordering above T_2 in the bulk Mn_5Si_3 crystals that have the same structure motif of the arrangement of Mn sites as our thin-film Mn_5Si_3 crystals. (The structural differences stemming from the locking of the thin film to the substrate, that otherwise play an important role in our study, do not affect the absence of the geometric frustration.) A non-collinear magnetic ordering is, therefore, a highly unlikely mechanism for the T-symmetry breaking that could generate our observed sizable spontaneous AHE accompanied by a vanishingly small net magnetization in the Mn_5Si_3 thin films. An alternative mechanism based on a sizable spontaneous canting of Mn spins (of unknown origin), that could generate a correspondingly sizable AHE, is also highly unfavorable because of the measured vanishingly small net magnetization.

(ii) To date, the only non-relativistic, therefore potentially comparably strong to ferromagnets, mechanism of T-symmetry breaking in the electronic structure of crystals with antiparallel ordering of spins and crystal-symmetry-driven vanishing magnetization has been identified in the class dubbed altermagnetic. This is an emerging class of collinear magnets, rigorously delimited based on a non-relativistic spin symmetry-group formalism,

with several material candidates theoretically predicted to date. (A recent review is in Ref. [15] of our manuscript.) The T-symmetry breaking in the electronic structure by the strong non-relativistic exchange mechanism, accompanied by a zero non-relativistic net magnetization, makes the altermagnetic ordering a highly favorable scenario for explaining the observed sizable spontaneous AHE accompanied by a vanishingly small net magnetization in our thin-film Mn_5Si_3 . More specifically, the d-wave type of the altermagnetic order is fully consistent with the set of our experimental and theoretical observations.

As the next step in our manuscript, to quantitatively compare the measured spontaneous AHE conductivity with ab initio theory, we choose the plausible microscopic AHE mechanism. For collinear magnets, symmetry dictates that relativistic spin-orbit coupling has to accompany the T-symmetry breaking in the electronic structure to allow for the AHE. Extensive earlier studies of relativistic microscopic AHE mechanisms in ferromagnets then suggest that for the longitudinal conductivities corresponding to the conductivities of our films, the Berry-curvature mechanism dominates the skew scattering mechanism. This explains why our microscopic AHE calculations are based on the relativistic Berry-curvature mechanism. Finally, we note that accounting for the relativistic spin-orbit coupling brings additional symmetry constraints besides the T-symmetry breaking on the presence of the AHE that, following the Reviewer's suggestion, we now explicitly summarize in the revised manuscript using the relativistic magnetic-group formalism.

After highlighting in the above general remarks the novelty of our work, and explaining the hierarchy of physics involved in our study, starting from the non-relativistic exchange and followed by the relativistic spin-orbit coupling, we now proceed to the response to specific comments and suggestions of the Reviewer.

Comment #1

1. The authors kept emphasizing that Mn_5Si_3 is perfectly compensated with precisely zero net magnetization. This is misleading. Although the strength of AHE and the magnitude of the net magnetization are not coupled, the AHE and the net magnetization have the same symmetry constraints, i.e. an AHE and a net magnetization (though it might be vanishingly small) must exist simultaneously. Therefore, a fully compensated antiferromagnets cannot host AHE. An antiferromagnet with AHE must be uncompensated, and thus can be seen as a "weak ferromagnet". In Fig.3a, it can be seen that the remanent net magnetization has the similar magnitude as that of Mn_3Sn (Ref. 18). Without this net magnetization, one cannot use a magnetic field to reverse the moments and hence the AHE, as shown in Fig.3 b,c. I suggest the authors revise their manuscript to avoid these misleading arguments. They also need to discuss the direction of the net magnetization, and the mechanism to generate the magnetization (Some canting in the presence of SOC? Some symmetry operation is

breaking?), since the Hall pseudovector has the same symmetry constraint as the net magnetization.

Response

The only place in the original manuscript where we mention “precisely zero net spontaneous magnetization” is in the discussion of Fig. 4a, where we state: “The rotation symmetries protect the compensated nature of the magnetic phase, i.e. the precisely zero net spontaneous magnetization in the non-relativistic limit,…” This statement is correct.

In cases not referring to the non-relativistic limit, we consistently used in the original manuscript a term “vanishing” when referring to the net magnetization. To make sure that there is no confusion, and following the Reviewer’s suggestion, we use in the revised text in these cases a term “vanishingly small”. In addition, we have also modified the 4th sentence of the first bold paragraph as follows:

“Recently, however, there have been predictions of an anisotropic d-wave magnetic phase that is robust and features an unconventional and attractive combination of a strong T-symmetry breaking in the electronic structure and a zero or only weak-relativistic magnetization [1, 8–15].”

The relativistic magnetic symmetries that do or do not allow for the weak magnetization and AHE, depending on the crystal direction of the Néel vector, are now explicitly discussed in the revised manuscript. We describe these changes in more detail in responses to the other Comments of the Reviewer.

Comment #2

2. In the main text, the authors confirmed that the crystal structure of the thin film is the same as that in bulk. And in the beginning of the theory part, they showed that the bulk is an antiferromagnet with the magnetic structure shown in Fig. 1a, and the thin film is “not” antiferromagnetic but a “d-wave magnet” with the magnetic structure shown in Fig. 1b, where the outer Mn-moments are reversed and the inner Mn-moments remain the same compared to these in Fig. 1a. The authors have not shown why the Fig. 1a in bulk changes to Fig. 1b in the thin film, the latter seems to be artificially proposed without any experimental evidence. Theoretically, the authors only compare the calculated energies of Fig. 1b and a ferromagnetic and a nonmagnetic phase, but have not shown the energy of Fig. 1a. Therefore, it’s not clear whether the Fig. 1b is stable compared to Fig. 1a.

Moreover, why can the high temperature magnetic order above 100 K be Fig. 1a with some small canting? Or a noncollinear antiferromagnetic order?

Response

We have partly addressed this point already above in the response to the General comment of the Reviewer. Here we address additional specific points.

In the original manuscript we stated that our experiments confirmed the same crystal-structure motif, not the overall crystal structure. The crystal-structure motif is described in the original manuscript as follows:

“The crystal-structure motif of Mn_5Si_3 , shown in Figs. 1d,e, is characterized by a distorted octahedron $[\text{Mn1Si}_6]$ with Si occupying its vertices and Mn1 in the center, and a distorted octahedron $[\square(\text{Mn2})_6]$ with Mn2 at the vertices and no atoms in its interior [25]. Since the distances of Mn atoms in pairs Mn1-Mn1, Mn1-Mn2 and Mn2-Mn2 are substantially different [25], the exchange interactions between Mn atoms do not exhibit the canonical geometric frustration [27].”

In contrast, the overall crystal structure of our thin films is not the same as that in bulk, and the differences are significant. The differences stem from the epitaxial strain and the epitaxial constraints. Apart from the different size of the lattice constants, the thin films do not undergo the orthorhombic and monoclinic crystal distortions that accompany the magnetic phase transitions at T_1 and T_2 in the bulk crystals. This was elaborated on in the original manuscript in the discussion of Fig. 2. Following the Reviewer’s comment, we have further highlighted the differences between bulk crystals and thin films in the revised text at the end of the discussion of Fig. 1 as follows:

“...They confirm that our thin films have the same crystal-structure motif as previously observed in the bulk samples. Apart from the same crystal-structure motif, there are important differences between the overall crystal structure of the bulk and our thin-film samples that stem from the epitaxial strain and the epitaxial constraints. The Mn_5Si_3 epilayers on the Si(111) substrate are constrained to a hexagonal crystal lattice in the whole studied temperature range and, therefore, the films do not undergo the structural transitions observed in bulk. In the following paragraphs, we elaborate on this point in more detail.”

In addition, we have revised the discussion of Fig. 2 as follows:

“We now compare the established temperature-dependent phenomenology in bulk Mn_5Si_3 to our measurements in the thin-film epilayers. As expected, the in-plane lattice parameters a and b of our epilayers, constrained by the substrate, show no transitions (Fig. 2a), and their weak temperature dependence closely follows the weakly decreasing in-plane lattice parameter with decreasing temperature of the Si substrate. In contrast, the out-of-plane

lattice parameter c of the Mn_5Si_3 film is not fixed by the substrate, and we observe an anomaly analogous to the T_2 transition observed in the bulk samples (Fig. 2b).

Note that in the case of Mn_5Si_3 on Si(111), the value of the in-plane lattice constant is governed primarily by the mismatch in the thermal expansion coefficients of the epilayer and the substrate. During cooling after growth, the mismatch in the thermal expansion coefficients, which are around $2.6 \times 10^{-6} \text{ K}^{-1}$ and $23 \times 10^{-6} \text{ K}^{-1}$ in Si and Mn_5Si_3 , respectively, causes an in-plane tensile strain. At room temperature and below we, therefore, find the in-plane lattice constant in our epilayers to be considerably larger than the bulk value. Consistently, the out-of-plane lattice constant in the epilayers is smaller than in bulk Mn_5Si_3 . In contrast to the thermal-expansion mismatch, the nominal mismatch of 3.7% between room-temperature in-plane lattice constants of the separate Si(111) and $\text{Mn}_5\text{Si}_3(0001)$ crystals plays a more minor role as it is partially accommodated by a thin MnSi interfacial layer between the Si substrate and the Mn_5Si_3 epilayer (see Methods for more details)."

The justification for focusing on the d-wave antiferromagnetic ordering as a favorable scenario, and the explanation why the alternative scenarios are unfavorable, is given above in our response to the General comment (as well as in the manuscript). In addition, to address this specific comment, we have added the following paragraph in the main-text discussion of Fig. 3 and the corresponding new Supplementary Fig. S5 and S6 containing the measured magnetoresistances:

"In Supplementary Fig. S5 and S6, we compare the field-dependence of the AHE with the longitudinal magnetoresistance. A strong negative magnetoresistance is observed below T_2 , consistent with the presence of the ρ^{T_H} contribution to the AHE that has been associated with the deviation of the magnetic order from the fully collinear state. Above T_2 where the ρ^{T_H} contribution is absent and the magnetic order is expected to be collinear, we observe the correspondingly weaker magnetoresistance."

In our DFT total-energy calculations we do not consider a comparison to the bulk antiferromagnetic state because, as emphasized above and in the manuscript, the bulk crystals have different lattice constants and are orthorhombically distorted in the antiferromagnetic phase. Moreover, the antiferromagnetic state does not allow for the AHE. We performed the conventional stability test by comparing the total energies of the paramagnetic and magnetic (in our case d-wave antiferromagnetic) phases, assuming in both phases the experimentally determined crystal structure and lattice parameters in our thin films. Only as an additional confirmation, we checked the comparison to the total energy of a hypothetical ferromagnetic phase assuming the same thin-film crystal-structure parameters.

Comment #3

3. Is there any special reason to plot the Brillouin zones with different centers? This makes the Fermi surface shown in Fig. 1 and Fig. 3 look different and results in confusions.

Response

Following the Reviewer's comment we extended the Fermi-surface plot in revised Fig. 1 to include the band structure around both the Γ -point and the M-point.

Comment #4

4. The authors described the nonrelativistic spin splitting of the band structure in Mn₅Si₃. However, it is not clear how the nonrelativistic spin splitting influences the Berry curvature and hence the AHE.

Response

In our response to the General comment of the Reviewer we have explained the hierarchy starting from non-relativistic exchange physics, that governs the strong T-symmetry breaking in the electronic structure, followed by relativistic spin-orbit coupling that in collinear magnets is an additional requirement for the AHE. In the manuscript, we have revised the two introductory paragraphs related to this point as follows:

“However, over the past decade, two types of crystal structures were predicted to host the spontaneous T-symmetry breaking responses, including a spontaneous AHE, that are not related to a net internal magnetization of the crystal [8, 16]: (i) The first type are geometrically frustrated structures, such as kagome, pyrochlore, or triangular lattices [17–19], where the experimentally observed spontaneous AHE [17–19] was related to a non-collinear magnetic ordering [18] or a spin-liquid state candidate [19].

(ii) For the second type of crystals, the distinctive feature are non-relativistic spin symmetries where the opposite-spin sublattices are connected by real-space rotation transformations and not by translation or inversion [1, 8, 11, 15, 20]. The spontaneous anomalous Hall response has then been related to a compensated collinear magnetic order with a vanishingly small (zero non-relativistic) magnetization [1, 8, 11, 15]. The general characteristic of the unconventional magnetism in this second type of crystals is a strong T-symmetry breaking and alternating spin polarization in both real-space crystal structure and momentum-space electronic structure, with or without the presence of the weak relativistic magnetization [11, 15]. The alternating spin polarization has suggested to refer to this phase as altermagnetism [11, 15]. Note that, in general, the T-symmetry breaking responses in altermagnets do not require relativistic spin-orbit coupling [11,15]. In the specific case of the AHE, however, additional symmetry breaking by the spin-orbit coupling is required in collinear magnets, including altermagnets [1,8,15].“

In addition, in the last paragraph of the introductory section, we have revised the text as follows:

“Our first-principles calculations show that without strong correlations, the unconventional d-wave magnetism of these magnetically ordered Mn atoms in the direct real space (Fig. 1b), and the corresponding d-wave spin polarization in the reciprocal momentum space (Fig. 1c), generate a vanishingly small net magnetization and a sizable spontaneous anomalous Hall conductivity of the microscopic Berry-curvature mechanism [1], consistent with our measurements...”

Comment #5

5. The magnetic order in Fig. 1b supports a C_{2x} rotation, which forbids AHE. This is contradictory to their results.

Response

Following this comment to avoid potential confusion, we have removed the C_{2x} rotation symmetry from the revised Fig. 1b and replaced it by highlighting the absence of the tT symmetry. The C_{2x} and C_{2y} symmetries are included in Fig. 4a which explicitly refers to the non-relativistic limit (spin-orbit coupling turned off). In addition, in the theory section, we have revised the discussion of Figs. 1 and 4 as follows:

“In real space, the candidate magnetic ordering shows the defining characteristics of the unconventional phase, dubbed altermagnetic: Namely the lack of translation or inversion and, in the non-relativistic limit, the presence of rotation symmetry transformations connecting opposite-spin sublattices. The rotation symmetries protect the compensated nature of the magnetic phase, i.e. the precisely zero net spontaneous magnetization in the non-relativistic limit, while allowing for the T-symmetry breaking and alternating spin splitting in the band structure [11, 15].”

Comment #6

6. The RuO₂ is also considered to be a “d-wave” by some of the authors in their recent works. In this sense, why are the AHE of RuO₂ and Mn₅Si₃ different, since the AHE is closely linked to the “d-wave” magnetism by the authors?

Response

In both RuO₂ and our thin-film Mn₅Si₃, the T-symmetry breaking in the electronic structure - the necessary condition for observing the AHE, is linked to the d-wave altermagnetic order. However, as explained in the discussion section of our manuscript, the d-wave altermagnetism in our thin-film Mn₅Si₃ is generated by the crystal positions in the hexagonal unit cell of the magnetic Mn atoms alone. In contrast, RuO₂ would be a conventional

antiferromagnet with opposite-spin sublattice connected by translation (and inversion), without the additional crystal-symmetry breaking by the arrangement of the non-magnetic (O) atoms in the unit cell.

An additional difference is linked to the relativistic spin-orbit coupling. As highlighted in the Reviewer's comments, and in our manuscript and responses to previous comments, the relativistic spin-orbit coupling in the electronic structure is required, in addition to the T-symmetry breaking, to allow for the AHE. As a result, as again emphasized in the comments by the Reviewer and now also explicitly in our revised manuscript, the relativistic magnetic symmetries determine whether the AHE is allowed or excluded for a given crystal-direction of the Néel vector. In RuO₂, the Néel-vector easy axis happens to be along the singular crystal direction for which the relativistic magnetic symmetries exclude the AHE. This is why, as discussed in detail in Refs. [8,10] of our manuscript, the spontaneous AHE is not observed in RuO₂, and for observing the AHE a magnetic field was applied to reorient the Néel vector away from the easy axis. In contrast, we observe the spontaneous AHE at zero field in our thin-film Mn₅Si₃.

Comment #7

7. The authors showed the calculation of AHC is performed with a [111] oriented Neel vector. Can this anisotropy be confirmed by experiment?

Response

To address this point we have revised the last two paragraphs of the theory section as follows:

“In Fig. 4f, we plot the DFT AHE conductivity as a function of the position of the Fermi level. The calculations are performed for the magnetic order vector pointing along the crystal direction [2-201] ([111] in the 3-component *a – b – c* notation)) between the in-plane [2-200] and normal-to-the-plane [0001] crystal axes. This off high-symmetry direction is chosen because it gives in our DFT calculations a lower total energy than the in-plane or normal-to-the-plane axes (see Supplementary information Sec. IV). Moreover, the magnetic point group -1 for the Néel vector along the [2-201] direction allows for a spontaneous anomalous Hall vector component along the [0001] crystal axis, i.e. along the normal to the thin-film plane, which makes it detectable in our experimental geometry. In contrast, AHE is excluded by symmetry in the magnetic point group *mmm* which corresponds in our case to the theoretically identified [0001] hard axis of the Néel vector. Also consistently with our measurements and DFT calculations, no spontaneous AHE would be detected for the Néel vector within the (0001)-plane (*c*-plane), (2-1-10)-plane (*a*-plane) or (0-110)-plane (*b*-plane) because in these cases the Hall vector, if allowed, is constrained by symmetry to the (0001)-plane of the thin film.

Our calculations in Fig. 4f illustrate that the spontaneous AHE conductivity, arising from the strong T-symmetry breaking in the electronic structure by the compensated collinear magnetic order of the unconventional d-wave phase, combined with the relativistic Berry-curvature mechanism, can reach values comparable to the AHE in common ferromagnets [4]. We obtain sizable $\sigma_H^U \approx 5 - 20 \text{ Scm}^{-1}$ within a $\sim 100 \text{ meV}$ energy window around the Fermi level. These theoretical values are consistent with our measurements.“

We note that complementary magnetization (SQUID) measurements of the magnetic anisotropy are inaccessible due to the vanishingly small moment in our thin (12 nm) Mn_5Si_3 films.

Comment #8

8. What is the direction of the Neel vector used when calculating the Fermi surface with SOC? What is the spin component shown in the Fermi surface with SOC? If the Neel vector is the low symmetric [111] direction, why are the Fermi surface and the spin distributions so symmetric/antisymmetric?

Response

We use the same Néel vector direction as in the AHE calculations. We specify the directions of the Néel vector and of the spin component in the revised captions of Fig. 1 and Fig. 4 as follows:

“The Néel vector is along the [2-201] crystal direction ([111] direction in the 3-component $a - b - c$ notation), and we plot spin projection on the [2-1-10] x-axis ([100] a-axis).”

The other spin projections give similar pictures. The seemingly symmetric spin distributions are another illustration that spin-orbit coupling generates only a weak distortion of the perfect non-relativistic d-wave symmetry, as emphasized in the following revised discussion of Fig. 4a,b:

“By comparing Fig. 4a and Fig. 4b, we see that the relativistic spin-orbit coupling generates only a weak perturbative correction in the Mn_5Si_3 Fermi surfaces. The d-wave form is preserved, and only the discrete 180° spin reversals when passing through the non-relativistic spin-degenerate nodal planes are replaced in the presence of the spin-orbit coupling by a continuous 180° spin reorientation. Note that in the relativistic calculations we considered the magnetic order vector pointing along the [2-201] crystal direction (for more details see the discussion below on the DFT AHE calculations and Supplementary information Sec. IV).”

Comment #9

9. Usually, the magnetism of a material is determined by the magnitudes and alignment of the magnetic moments. In this sense, the Mn_5Si_3 should be an antiferromagnet. In this work, however, the authors proposed the concept “d-wave magnetism” according to the nonrelativistic electronic structure. I think such a concept is unnecessary and may lead to many confusions. For example, everyone in the community of magnetic materials and magnetism will say both Fig. 1a and Fig. 1b are antiferromagnetic, but the current work seems to suggest Fig. 1b is not antiferromagnetic.

Response

In the response to the General comment and other comments of the Reviewer, and in the revised manuscript, we have highlighted the importance of the hierarchy starting from non-relativistic exchange physics that governs the strong T-symmetry breaking. Apart from AHE, this T-symmetry breaking generates numerous other phenomena that can be also of non-relativistic nature, and that for many decades have been broadly considered to be excluded in the entire class commonly called, in line with the Reviewer’s comment, collinear antiferromagnets. Since recently, however, it has been realized that this, for many decades established, notion does not apply to a range of crystals traditionally called collinear antiferromagnets. Instead, they have been demonstrated or predicted to exhibit phenomena such as the giant and tunneling magnetoresistance and spin-transfer torque, analogous to those in ferromagnets, or the alternating spin-polarization in the electronic structure, transverse pure spin currents or chirality-split magnons, with no counterparts in the traditional ferromagnetic/antiferromagnetic phenomenology.

A mathematically rigorous classification based on the spin-group formalism has then established that all collinear magnets split into three distinct spin-group types. The first and second ones correspond to the traditional notion and phenomenology of collinear ferromagnets and antiferromagnets, while the third type describes compensated collinear magnets with the recently identified unconventional phenomenology. It is not the particular terminology used to refer to this third class that principally matters here. What matters physically is its exclusively distinct symmetry nature and unconventional phenomenology. A new terminology is introduced as a reminder about this emerging unconventional physics. These points are thoroughly discussed in the review article in Ref. [15] of our manuscript.

Summary comment

In summary, in this work, the authors have reported an AHE in Mn_5Si_3 . They claimed that this phenomenon is attributed to the artificially proposed magnetic order in Fig. 1b, which does not have experimental and theoretical evidence. The authors focused on discussing the nonrelativistic electronic structures induced by the magnetic order in Fig. 1b and proposed a new concept “d-wave magnetism”. However, it’s not clear why this nonrelativistic concept is important for the AHE, since AHE is a relativistic phenomenon. I suggest the authors perform additional measurements as possible for the magnetic

structure and magnetic anisotropy, and try to explain their observations using conventional languages such as magnetic space/point groups, the symmetry allowed net magnetization and the conductivity tensors, the Berry curvatures, etc.

Response

In our responses to the previous Reviewer's comments, and in the revised manuscript, we highlight that the magnetic order we consider is not artificially proposed but, in contrast to competing scenarios, provides an interpretation consistent with the full set of experimental and theoretical observations. We have also highlighted the hierarchy starting from non-relativistic exchange physics, that governs the strong T-symmetry breaking in the electronic structure, followed by relativistic spin-orbit coupling that in collinear magnets is an additional requirement for the AHE. Accounting for the relativistic spin-orbit coupling brings additional symmetry constraints on the presence of the AHE besides the T-symmetry breaking that, following the Reviewer's comments, we now explicitly summarize in the revised manuscript using the relativistic magnetic-group formalism.

REVIEWER COMMENTS

Reviewer #1 (Remarks to the Author):

The authors have addressed most of my concerns. Basically the quality of the work is quite good, because the present work provides a new material to show the anomalous Hall effect, and the authors have conducted extensive work to show the correlation between the microstructure/electronic structure and the anomalous Hall effect. The anomalous Hall effect in antiferromagnetic Mn₅Si₃ is quite different from the case in other antiferromagnetic materials. From my opinion, the work is very close to the acceptance. There are only two tiny issues should be addressed.

1. The unit of magnetization of Mn₅Si₃ and Si substrate seems incorrect (Fig. 3 and Fig. S3). $1 \text{ A/m} = 0.001 \text{ emu/cc}$. If the size of the film for SQUID measurement is around $10 \text{ nm} \times 5 \text{ mm} \times 5 \text{ mm}$, the magnetization of around 10 A/m at 5 T corresponds to $2.5 \times 10^{-10} \text{ emu}$. However, the detection limit of SQUID is 10^{-8} emu . The author should also present the calculation process for $\text{m}\mu\text{B/u.c.}$, which seems inconsistent with their SQUID results. Besides, the saturated magnetization of MnSi ($5 \text{ kA/m} = 5 \text{ emu/cc}$) as presented in Fig. S10b is also questionable. Typical saturated magnetization of ferromagnets is around 1000 emu/cc .

2. The authors may misunderstand my previous 5# comment. In previously published papers, the easy axis of bulk Mn₅Si₃ is along in-plane b axis (PRB, 103, 024407 (2021)). Did the authors compare the MAE of [2-201] with b axis? Why the easy axis changes to this direction of low symmetry for thin film Mn₅Si₃? Does epitaxial strain bring about this change? For the space group of P6₃/mcm as proposed by the authors, there are several equivalent crystal orientations of [2-201], such as [20-21] or [02-21]. Does this mean thin film Mn₅Si₃ have several tilted easy axis? I am confused for Mn₅Si₃ thin film, if the orthorhombic crystal distortion at T₁ is missing, what drives the magnetic phase transition from paramagnetism to collinear AFM? In other words, for bulk Mn₅Si₃, collinear AFM emerges at 2/3 of Mn₂ atoms below T₁, which is reasonable because $a \neq b$ to break the symmetry. However, for thin film Mn₅Si₃ of P6₃/mcm with $a = b$, why only 2/3 of Mn₂ atoms become collinear AFM? Even 2/3 of Mn₂ atoms become magnetic, there should be 3 equivalent possibilities because the 3 triangular arrangements of Mn₂ at the height of $1/4c$ and $3/4c$ in a unit cell are indistinguishable from each other. I would appreciate the authors can give a clear schematic diagram of magnetic Mn atoms and their magnetization directions for better understanding. If this is indeed difficult from the experimental side, some comments may be also useful.

Reviewer #2 (Remarks to the Author):

The revised manuscript provided by the authors made several minor, but wholly adequate changes in response to my questions. As I stated in my earlier report, I feel this is a strong contribution to the emerging story of altermagnetism, and I feel this is very appropriate for Nature Communications. I feel the authors have also gone to great lengths to address the more substantive questions of other referees. I feel this has only strengthened the paper.

Reviewer #3 (Remarks to the Author):

In the response and the revision, the authors have emphasized that “the primary focus of our manuscript on the identification of the candidate magnetic phase with T-symmetry broken electronic structure”, which I agree. And that’s why, I, and other reviewers, such as reviewer one, are prudent to recommend, especially for such a high-profile journal, since it would be awkward if experiments in the future show the real magnetic order is not the one proposed in the current work.

The authors have analyzed that the noncollinear AFM order or bulk phase with canting are not possible. These analyses seem to be reasonable, but are not sufficient to identify the proposed one is the real magnetic order. Experimentally, although a single film gives the weak signal in neutron scattering, it’s well known that one can use multiple films to enhance the signal and get the magnetic structure. Theoretically, one should compare the energies of several candidate magnetic orders and pick the low energy one.

1. In the responses to reviewer one and to me, the authors have confirmed that there is a significant strain and an associated lattice variation. But how does the lattice variation transform Fig. 1a to Fig. 1b has still not been explained.

2.1 The total energy calculations in the current version can only confirm the magnetic order is neither ferromagnetic nor nonmagnetic. It cannot exclude that some other magnetic order, such as the Fig. 1a, has a much lower energy. And the authors refused to compare the energy of Fig. 1a and Fig. 1b because the associated lattice parameters are different, and Fig. 1a does not support AHE. While, if Fig. 1a indeed show a low energy compared to that in Fig. 1b, it does not mean we should adopt Fig. 1a as the ground state, but it means we should look for other magnetic order which hosts both the lower energy and AHE.

2.2 A conventional method is to calculate the energies of two magnetic orders in Fig. 1(a,b) (and some other possible magnetic orders) from the bulk structure toward the thin film structure. If it is as expected by the authors that the strain and lattice variation cause the two different magnetic order, the Fig. 1a should have lower energy in bulk structure, and the Fig. 1b should have the lower energy in the thin film structure, and there will be an energy crossover of these two orders in somewhere in the middle. This is a very convenient calculation and should be present.

3. I agree that “complementary magnetization (SQUID) measurements of the magnetic anisotropy are inaccessible due to the vanishingly small moment in our thin (12 nm) Mn₅Si₃ films.” But since it has such a strong AHE response, this anisotropy should be reflected in AHE measurement in PPMS. This should be also convenient, I think, as there is a relativistic net magnetization which can serve as a knob for the control of Neel vector. Indeed, they authors can reverse the AHE, which implies the Neel vector is reversed. With the same logic, the rotation of Neel vector can be also realized. The authors said “Also consistently with our measurements and DFT calculations, no spontaneous AHE would be detected for the Néel vector within the (0001)-plane (c-plane), (2⁻1⁻10)-plane (a-plane) or (0⁻110)-plane (bplane) because in these cases the Hall vector, if allowed, is constrained by symmetry to the (0001)-plane of the thin film”. I didn’t find these measured results in the revision.

4. I still would like to see the analysis on the distribution of Berry curvature on the FS, like the authors did in their nice work for AHE in RuO₂.

5. While, I understand the authors’ insistence on the new concept “altermagnets” or “d-wave magnet” they proposed, and agree they are interesting. But proposing a new terminology does not mean we need to negate traditional ones. These new and traditional terminologies can be compatible. For example, the noncollinear antiferromagnets also have spin-dependent transport properties with no counterparts in the traditional ferromagnetic/antiferromagnetic phenomenology, but they are well accepted as a subset of antiferromagnets. If the authors would be willing to describe the “altermagnets” as a subset of collinear antiferromagnets, the new terminology they proposed should be much easier to accept by the community. Otherwise, I don’t know what we should call the ferrimagnets in the future, which is well defined in the traditional classification and is a hot topic in the research fields of magnetism and spintronics, but has no room in the new classification the author proposed.

Reviewer #1

Comment #1

1. The unit of magnetization of Mn₅Si₃ and Si substrate seems incorrect (Fig. 3 and Fig. S3). $1 \text{ A/m} = 0.001 \text{ emu/cc}$. If the size of the film for SQUID measurement is around $10 \text{ nm} \times 5 \text{ mm} \times 5 \text{ mm}$, the magnetization of around 10 A/m at 5 T corresponds to $2.5 \times 10^{-10} \text{ emu}$. However, the detection limit of SQUID is 10^{-8} emu . The author should also present the calculation process for $m\mu\text{B}/\text{u.c.}$, which seems inconsistent with their SQUID results. Besides, the saturated magnetization of MnSi ($5 \text{ kA/m} = 5 \text{ emu/cc}$) as presented in Fig. S10b is also questionable. Typical saturated magnetization of ferromagnets is around 1000 emu/cc .

Response

The plotted values were obtained as follows:

Si only sample (Fig.S3): size $0.5\text{mm} \times 5\text{mm} \times 5\text{mm}$, measured signal $\sim 100 \times 10^{-6} \text{ emu}$. The magnetization is then $M \sim 10 \times 10^{-3} \text{ emu/cc} = 10 \text{ A/m}$, as indicated in Supplementary Fig. S3.

Sample with Mn₅Si₃ (Fig. 3): The plot shows the total magnetization of the measured sample, i.e., also including the substrate. The inset also shows the total measured magnetization. However, when recalculating from A/m to $m\mu\text{B}/\text{u.c.}$, we considered that the remanent signal at zero field is due to the Mn₅Si₃ film alone and has no contribution from the Si substrate (we also considered the same for the small field range around zero plotted in the inset). **These points are now mentioned explicitly in the caption of Fig. 3.**

As stated in the manuscript, the aim of Fig. 3 is to highlight the absence of a sizable remanent moment, i.e., the absence of a correlation between the measured magnetization and the AHE.

MnSi reference sample. The moment of MnSi is indeed anomalously low as compared to conventional ferromagnets. Our measured value is of the same order of magnitude as reported in earlier literature on MnSi (e.g. <https://link.springer.com/article/10.1186/s11671-020-03462-2>). We have included the above reference in Methods after the sentence “In Supplementary Sec. III we summarize measurements on a control thin epitaxial film of MnSi deposited on Si(111) [43].”

Comment #2

2. The authors may misunderstand my previous 5# comment. In previously published papers, the easy axis of bulk Mn₅Si₃ is along in-plane b axis (PRB, 103, 024407 (2021)). Did the authors compare the MAE of [2-201] with b axis? Why the easy axis changes to this direction of low symmetry for thin film Mn₅Si₃? Does epitaxial strain beings about this change? For the space group of P6₃/mcm as proposed by the authors, there are several equivalent crystal orientations of [2-201], such as [20-21] or [02-21]. Does this mean thin film Mn₅Si₃ have several tilted easy axis? I am confused for Mn₅Si₃ thin film, if the orthorhombic crystal distortion at T₁ is missing, what drives the magnetic phase transition from paramagnetism to collinear AFM? In other words, for bulk Mn₅Si₃, collinear AFM emerges at 2/3 of Mn₂ atoms below T₁, which is reasonable because $a \neq b$ to break the symmetry. However, for thin film Mn₅Si₃ of P6₃/mcm with $a = b$, why only 2/3 of Mn₂ atoms become collinear AFM? Even 2/3 of Mn₂ atoms become magnetic, there should be 3 equivalent possibilities because the 3 triangular arrangements of Mn₂ at the height of $1/4c$ and $3/4c$ in a unit cell are indistinguishable from each other. I would appreciate the authors can give a clear schematic diagram of magnetic Mn atoms and their magnetization directions for better understanding. If this is indeed difficult from the experimental side, some comments may be also useful.

Response

In our DFT calculations for the altermagnetic phase, the body-diagonal direction of the Néel vector gives a lower energy than the in-plane or normal-to-the-plane directions. As explained in the manuscript, this is consistent with the experimentally observed AHE signal at remanence. We agree with the Reviewer that because only four out of six Mn₂ atoms are magnetic, there are three crystallographically equivalent possibilities for the distribution of these magnetic Mn₂ atoms (three equivalent Mn₂ “quadruplets”). This in effect can restore the hexagonal symmetry of the magnetic responses, including the AHE. To test this, we have performed additional measurements of the AHE in a strong saturating (4 T) magnetic field rotated in the (2-1-10) plane and the two other crystallographically equivalent planes, and in the (0-110) plane and the two other crystallographically equivalent planes. As expected, the AHE traces for field rotations in crystallographically equivalent planes fall on top of each other, while there is a clear distinction between the non-equivalent planes. We also note that these AHE rotation traces show a strong anisotropy of the AHE, i.e., a strong deviation from a $\cos\theta$ dependence, where θ is the field-rotation angle from normal-to-the-

plane to in-plane direction. In contrast, the measured weak magnetization induced by the rotated magnetic fields is isotropic, i.e., its out-of-plane component follows the $\cos\theta$ dependence. This provides additional confirmation that the AHE is not correlated to the weak magnetization.

The new AHE measurements are mentioned in the main text after the discussion of Fig.~3 and are shown in the revised Supplementary information in Fig. S9 with the above description included in the figure caption.

Reviewer #3

Comment #1

1. In the responses to reviewer one and to me, the authors have confirmed that there is a significant strain and an associated lattice variation. But how does the lattice variation transform Fig. 1a to Fig. 1b has still not been explained.

Response

As recalled in our manuscript, the unconstrained bulk undergoes a structural transition at T_1 , that lifts the degeneracy between the in-plane a and b lattice parameters, accompanied by a magnetic transition into an antiferromagnetic state that doubled the unit cell along the b -axis. The tT -symmetry of this antiferromagnetic state excludes the AHE, consistent with experiments in the unconstrained bulk samples. In contrast, we elaborate in the manuscript on our measurements in the epilayers showing the absence of the structural transition. This is consistent with the expectation that the Mn_5Si_3 epilayers on the Si(111) substrate are constrained to a hexagonal crystal lattice in the whole studied temperature range and, therefore, the in-plane lattice parameters a and b of our epilayers, constrained by the substrate, show no transition. We also note in the manuscript that in the case of our Mn_5Si_3 epilayers on Si(111), the value of the in-plane lattice constant is governed primarily by the mismatch in the thermal expansion coefficients of the epilayer and the substrate. In contrast to the thermal-expansion mismatch, the nominal mismatch between room-temperature in-plane lattice constants of the separate Si(111) and Mn_5Si_3 (0001) crystals plays a more minor role as it is partially accommodated by a thin MnSi interfacial layer between the Si substrate and the Mn_5Si_3 epilayer.

Regarding the magnetic transition, we emphasize in the manuscript that the observation of the AHE in our epilayers below T_1 shows that the antiferromagnetic state with the unit cell doubled along the b -axis, that in the unconstrained bulk accompanies the structural transition with different a and b lattice parameters, does not form in our epilayers. Instead, the proposed antiferromagnetic phase is consistent with both the structural and AHE measurements in our substrate-constrained Mn_5Si_3 epilayers.

To further test our interpretation, we have performed additional measurements of the AHE in a strong saturating (4 T) magnetic field rotated in the (2-1-10) plane and the two other crystallographically equivalent planes of the hexagonal unit cell, and in the (0-110) plane and the two other crystallographically equivalent planes. As expected, the AHE traces for field rotations in equivalent planes fall on top of each other, while there is a clear distinction between the non-equivalent planes. Note that because only four out of six Mn₂ atoms are magnetic below T₁, there are three crystallographically equivalent possibilities for the distribution of these magnetic Mn₂ atoms in the considered altermagnetic phase (three equivalent Mn₂ “quadruplets”). This in effect can restore the hexagonal symmetry of the magnetic responses, including the AHE, as confirmed in the new field-rotation AHE measurements included in the revised manuscript. We also note that these AHE rotation traces show a strong anisotropy of the AHE, i.e., a strong deviation from a $\cos\theta$ dependence, where θ is the field-rotation angle from normal-to-the-plane to in-plane direction. In contrast, the measured weak magnetization induced by the rotated magnetic fields is isotropic, i.e., its out-of-plane component follows the $\cos\theta$ dependence. This provides additional confirmation that the AHE is not generated by the weak magnetization.

The new AHE measurements are mentioned in the main text after the discussion of Fig.~3 and are shown in the revised Supplementary information in Fig. S9 with the above description included in the figure caption.

Comment #2

2.1 The total energy calculations in the current version can only confirm the magnetic order is neither ferromagnetic nor nonmagnetic. It cannot exclude that some other magnetic order, such as the Fig. 1a, has a much lower energy. And the authors refused to compare the energy of Fig. 1a and Fig. 1b because the associated lattice parameters are different, and Fig. 1a does not support AHE. While, if Fig. 1a indeed show a low energy compared to that in Fig. 1b, it does not mean we should adopt Fig. 1a as the ground state, but it means we should look for other magnetic order which hosts both the lower energy and AHE.

2.2 A conventional method is to calculate the energies of two magnetic orders in Fig. 1(a,b) (and some other possible magnetic orders) from the bulk structure toward the thin film structure. If it is as expected by the authors that the strain and lattice variation cause the two different magnetic order, the Fig. 1a should have lower energy in bulk structure, and the Fig. 1b should have the lower energy in the thin film structure, and there will be an energy crossover of these two orders in somewhere in the middle. This is a very convenient calculation and should be present.

Response

We have not considered other possible magnetic phases in our DFT calculations for the following two reasons, highlighted in the manuscript: (i) The altermagnetic phase represents

a minimal modification compared to the antiferromagnetic phase that was established in the unconstrained bulk materials – the two phases share the same antiparallel arrangement of the four magnetic Mn₂ atoms, and the altermagnetic phases differs from the antiferromagnetic phase only in the absence of the doubling of the unit cell. This minimal modification supports the plausibility of our altermagnetic interpretation. (ii) Since the distances of Mn atoms in pairs Mn₁–Mn₁, Mn₁–Mn₂ and Mn₂–Mn₂ are substantially different, the exchange interactions between Mn atoms do not exhibit the canonical geometric frustration. Therefore, the earlier established scenario of the AHE accompanied by a vanishing magnetization in frustrated lattices, such as the kagome lattice, does not apply in our case. We are not aware from literature of any other magnetic-ordering scenarios that would generate a strong remanent AHE accompanied by a vanishing magnetization.

We have not performed the systematic DFT comparison between the antiferromagnetic and altermagnetic phases because our Mn₅Si₃ epilayers are not free-standing thin films and, therefore, they do not differ from the bulk crystals merely by the lattice parameters. The crystal and magnetic structure of our Mn₅Si₃ thin films is intimately related to the constraints imposed by the Si substrate and the MnSi interfacial layer, as evidenced by the experiments described in our manuscript. Since molecular-beam epitaxy is a non-equilibrium growth method, and since the multiple interfaces inevitably involve a formation of a range of defects, a reliable DFT modelling of all these interfacial complexities is practically impossible. (In general, there is no existing ab initio based description of the molecular beam epitaxy). However, these complex interfacial effects are likely to affect the energetics significantly more than the mere energy difference between the double-unit-cell antiferromagnetic and the single-unit-cell altermagnetic phase in a hypothetical free-standing crystal.

Comment #3

3. I agree that “complementary magnetization (SQUID) measurements of the magnetic anisotropy are inaccessible due to the vanishingly small moment in our thin (12 nm) Mn₅Si₃ films.” But since it has such a strong AHE response, this anisotropy should be reflected in AHE measurement in PPMS. This should be also convenient, I think, as there is a relativistic net magnetization which can serve as a knob for the control of Neel vector. Indeed, they authors can reverse the AHE, which implies the Neel vector is reversed. With the same logic, the rotation of Neel vector can be also realized. The authors said “Also consistently with our measurements and DFT calculations, no spontaneous AHE would be detected for the Néel vector within the (0001)-plane (c-plane), (2̄ 1̄ 10)-plane (a-plane) or (0̄ 110)-plane (bplane) because in these cases the Hall vector, if allowed, is constrained by symmetry to the (0001)-plane of the thin film”. I didn’t find these measured results in the revision.

Response

Motivated also by this comment, we have performed the additional systematic field-rotation AHE measurements, mentioned in the response to Comment #1, whose results are consistent with our interpretation.

Comment #4

4. I still would like to see the analysis on the distribution of Berry curvature on the FS, like the authors did in their nice work for AHE in RuO₂.

Response

We have included the Berry curvature calculations in the revised Supplementary information in Fig. S13.

Comment #5

5. While, I understand the authors' insistence on the new concept "altermagnets" or "d-wave magnet" they proposed, and agree they are interesting. But proposing a new terminology does not mean we need to negate traditional ones. These new and traditional terminologies can be compatible. For example, the noncollinear antiferromagnets also have spin-dependent transport properties with no counterparts in the traditional ferromagnetic/antiferromagnetic phenomenology, but they are well accepted as a subset of antiferromagnets. If the authors would be willing to describe the "altermagnets" as a subset of collinear antiferromagnets, the new terminology they proposed should be much easier to accept by the community. Otherwise, I don't know what we should call the ferrimagnets in the future, which is well defined in the traditional classification and is a hot topic in the research fields of magnetism and spintronics, but has no room in the new classification the author proposed.

Response

We understand that the nomenclature of the different magnetic phases may be debated. However, we point out that in physics, one of the most common tools for a basic classification of phases of matter is symmetry. In our paper we refer to altermagnets as a distinct phase in this sense, i.e., from the basic symmetry-classification viewpoint, as derived in Refs. [11,15]. The spin-group symmetry classification of Refs. [11,15] focuses, within the hierarchy of interactions, on the strong non-relativistic exchange and on collinear spin arrangements on crystals. It classifies collinear magnets into three mutually exclusive symmetry classes termed in Refs. [11,15]: (i) conventional ferromagnets (ferrimagnets), (ii) conventional antiferromagnets, and (iii) altermagnets. The spin-group classification into the three mutually exclusive symmetry classes is mathematically rigorous, systematic and complete for all collinear spin arrangements on crystals, as derived in detail in Refs. [11,15]. Within the spin-group symmetry classification, referring to three mutually exclusive classes is, therefore, not a matter of our choice but a mathematically rigorous result.

To reflect the above points in the manuscript, we have revised the relevant introductory paragraph as follows:

“(ii) For the second type of crystals with a collinear magnetic order, termed altermagnetic[11, 15], the distinctive features are non-relativistic spin symmetries where the opposite-spin sublattices are connected by real-space rotation transformations and not by translation or inversion[1, 8, 11, 15, 20]. In contrast, conventional collinear ferromagnets (ferrimagnets) and antiferromagnets have exclusively distinct symmetries[11, 15]: ferromagnets (ferrimagnets) have only one spin lattice (or opposite-spin sublattices not connected by any symmetry transformation), and antiferromagnets have opposite-spin sublattices connected by a real-space translation or inversion. The spontaneous anomalous Hall response in altermagnets has then been related, when including relativistic spin-orbit coupling, to a compensated collinear magnetic order with a vanishingly small (zero non-relativistic) magnetization[1, 8, 11, 15]. The general characteristic of the unconventional magnetism in altermagnets is a strong \mathbb{T} -symmetry breaking and alternating spin polarization in both real-space crystal structure and momentum-space electronic structure, with or without the presence of the weak relativistic magnetization [11, 15]. Note that, in general, the \mathbb{T} -symmetry breaking responses in altermagnets do not require relativistic spin-orbit coupling[11, 15]. In the specific case of the AHE, however, additional symmetry breaking by the spin-orbit coupling is required in collinear magnets, including altermagnets[1, 8, 15].”

The above revised introductory paragraph includes an explicit remark on ferrimagnets, following the spin-group symmetry classification as derived in Refs. [11, 15]. (We note that a more detailed discussion of ferromagnets and ferrimagnets belonging into one basic symmetry class can be found in the editorial of Ref. 15 at <https://doi.org/10.1103/PhysRevX.12.040002>.) Consequently, ferrimagnetism is included in the spin-group symmetry classification.

Regarding non-collinear magnets, we note that a systematic symmetry classification tailored to the hierarchy of interactions in the non-collinear magnetic crystals is yet to be completed. However, some partial results of the classification have been already presented. For example, unlike the d-wave magnetic counterpart in collinear altermagnets of the d-wave unconventional superconductivity in cuprates, non-collinear compensated magnets can host unconventional p-wave counterparts of the p-wave superfluid He-3 (arXiv:2309.01607). These very recent developments contrast with the many decades of quantum magnetism research over which unconventional higher parity-wave forms of magnetism have remained theoretically and experimentally elusive, or were even considered principally excluded. Clearly there is now an emerging magnetism landscape that goes well beyond the conventional ferromagnetism (ferrimagnetism) and antiferromagnetism. How the terminology will adapt to these exciting new developments, including those in non-collinear magnets, is yet to be seen.

REVIEWER COMMENTS

Reviewer #1 (Remarks to the Author):

I am satisfied with the authors' response. The altermagnet is an emergent and interesting topic in spintronics community. Thus I recommend its publication as it is.

Reviewer #3 (Remarks to the Author):

I am still not convinced about the magnetic order the authors proposed, since no direct experimental or theoretical evidence supporting it. But I guess that it's difficult for the authors at this stage to completely prove that the magnetic order in their Mn₅Si₃ film is exactly the same as that shown in Fig. 1(b), and to explain why it occurs. So, I decide to lower the criteria and recommend this work, since the AHE, and the properties of the proposed magnetic order, are solely interesting. However, although the authors have called that in Fig. 1(b) a "candidate" magnetic order, the way the manuscript was written may mislead the readers to consider that the magnetic structure of Mn₅Si₃ film would be definitely the one shown in Fig. 1(b), which I am not that confident as the authors since the current results are not sufficient. Therefore, I urge the authors to naturalize their strong statements, and clearly emphasize there still requires efforts to be devoted to solving the unclear puzzles about the magnetic order. I list my suggestions below:

1. I suggest the authors add a paragraph to summon the efforts in the future: (1) A direct experimental detection of the magnetic order (though challenging) is required to verify whether the magnetic order of the Mn₅Si₃ is indeed that shown in Fig. 1b; (2) The mechanism why this magnetic order can occur should be established in the future. The authors can mention the complex factors associated with interfaces and seed layers here, and suggest some experiments or calculations helpful for solving this problem, such as comparing the samples with different substrate and with different thickness/stoichiometries of seed layers, etc.

2. I still suggest the authors calculate and directly show the total energy of the magnetic order in Fig. 1a using the lattice parameters they used in section IV of SM. I expect the calculation will show that the order in Fig. 1a has lower energy. But it is fine, since there are lots of materials exist in nature with the high energy metastable phases as long as they are in the local minimum. Knowing the energy difference between the orders of Figs. 1a and 1b for the hypothetical free-standing crystal will be helpful for the future computational studies of the mechanisms of the magnetic orders in the

Mn₅Si₃ film. For example, one can calculate the variation of this energy difference for Mn₅Si₃ with different interfacial conditions in the heterostructures, and check what condition can reduce or even reverse this energy difference.

3. In section V of SM, the authors claimed that the direct measurement of spin structure of Mn₅Si₃ cannot be conclusive. This argument seems to be dogmatic and inappropriate, as it expressed a negative attitude on the further experimental efforts on the detection of the magnetic structure. If the magnetism in Mn₅Si₃ is important as the authors emphasized, the magnetic order must be detected and confirmed by experiment. Otherwise, it will be just a hypothesis. As I mentioned in my last report, it's well known that one can use many pieces of the film samples to enhance the signal for deriving the magnetic structure. Also, one can detect the spin-split electronic structure by ARPES and compare it with that shown in this work. The authors should be positive and suggest potential experimental verifications of the magnetic order in this part.

Reviewer #1 (Remarks to the Author):

I am satisfied with the authors's response. The altermagnet is an emergent and interesting topic in spintronics community. Thus I recommend its publication as it is.

We thank the referee for recommending our work.

Reviewer #3 (Remarks to the Author):

I am still not convinced about the magnetic order the authors proposed, since no direct experimental or theoretical evidence supporting it. But I guess that it's difficult for the authors at this stage to completely prove that the magnetic order in their Mn₅Si₃ film is exactly the same as that shown in Fig. 1(b), and to explain why it occurs. So, I decide to lower the criteria and recommend this work, since the AHE, and the properties of the proposed magnetic order, are solely interesting.

We thank the referee for recommending our work.

However, although the authors have called that in Fig. 1(b) a “candidate” magnetic order, the way the manuscript was written may mislead the readers to consider that the magnetic structure of Mn₅Si₃ film would be definitely the one shown in Fig. 1(b), which I am not that confident as the authors since the current results are not sufficient. Therefore, I urge the authors to naturalize their strong statements, and clearly emphasize there still requires efforts to be devoted to solving the unclear puzzles about the magnetic order. I list my suggestions below:

1. I suggest the authors add a paragraph to summon the efforts in the future: (1) A direct experimental detection of the magnetic order (though challenging) is required to verify whether the magnetic order of the Mn₅Si₃ is indeed that shown in Fig. 1b;

We acknowledge that while the proposed magnetic order aligns with our experimental observations, future efforts should be focused on directly observing the spin structure. This is discussed in detail in the Supplementary Information, Section V. We revised the main text by stating that the experimental evidence for the proposed magnetic order is still missing to underscore the significance of forthcoming experiments in verifying the spin structure. We added a paragraph on possible next experiments in the main text and expanded Section V in the Supplementary Information where we provide more details on concrete techniques (changes are in red color in the main text and SI).

(2) The mechanism why this magnetic order can occur should be established in the future. The authors can mention the complex factors associated with interfaces and seed layers here, and suggest some experiments or calculations helpful for solving this problem, such as comparing the samples with different substrate and with different thickness/stoichiometries of seed layers, etc.

Our study introduces a new phase of Mn₅Si₃, which emerges due to epitaxial constraints imposed by the silicon substrate. This has implications beyond the observation of the anomalous Hall effect, opening a new exploratory phase in the study of this compound. The impact of our findings is demonstrated by a newly published paper on the electrical switching of Mn₅Si₃ thin films <https://www.science.org/doi/10.1126/sciadv.adn0479#supplementary-materials>. We provide more details in the Section V in the Supplementary Information and refer to it in the main text.

Our proposed future work includes the following directions:

- 1) Investigating the role of the substrate by experimenting with alternatives, such as GaAs, which is epitaxially compatible. Eventually test the role of buffer composition to tune the lattice matching/strain.
- 2) Studying the influence of strain, both by applying strain to the films and potentially to the bulk material. This could reveal whether altermagnetism can be modulated or toggled by external strain.
- 3) Conducting a stoichiometric series analysis of Mn₅Si_{1-x}Ge_x. Since Mn₅Ge₃ is a ferromagnetic compound, substituting Si for Ge alters the interatomic distances. It would be

intriguing to determine at what point the compound becomes compensated and altermagnetic and to thoroughly investigate the role of interatomic distance in this context.

2. I still suggest the authors calculate and directly show the total energy of the magnetic order in Fig. 1a using the lattice parameters they used in section IV of SM. I expect the calculation will show that the order in Fig. 1a has lower energy. But it is fine, since there are lots of materials exist in nature with the high energy metastable phases as long as they are in the local minimum. Knowing the energy difference between the orders of Figs. 1a and 1b for the hypothetical free-standing crystal will be helpful for the future computational studies of the mechanisms of the magnetic orders in the Mn₅Si₃ film. For example, one can calculate the variation of this energy difference for Mn₅Si₃ with different interfacial conditions in the heterostructures, and check what condition can reduce or even reverse this energy difference.

We agree with referee that for capturing realistic energy of the altermagnetic and antiferromagnetic phases more future calculations which would include interfacial effects are needed. Following referees suggestion we have computed the ground-state energies of the free standing Mn₅Si₃ in the altermagnetic and antiferromagnetic phase with the experimental strained lattice parameters and we have added following note to SI Section IV.

In the following table we summarize the ground-state energies for the two phases considered in the main text Fig. 1a and b with the strained lattice parameters and relaxed atomic positions.

	ENERGY	a/A	c/A
Altermagnet	-127,46	6,9500	4,7900
Antiferromagnet	-127,59	6,9500	4,7900

The resulting energy for the antiferromagnetic state is 0.13 eV per the unit cell lower than for the altermagnetic state. We point out that in reality altermagnetic state can be realized in local energy minimum and that also such a small energy difference can be overturned by interfacial effects in the heterostructure. It would be interesting to account for these effects in future computational studies.

3. In section V of SM, the authors claimed that the direct measurement of spin structure of Mn₅Si₃ cannot be conclusive. This argument seems to be dogmatic and inappropriate, as it expressed a negative attitude on the further experimental efforts on the detection of the magnetic structure. If the magnetism in Mn₅Si₃ is important as the authors emphasized, the magnetic order must be detected and confirmed by experiment. Otherwise, it will be just a hypothesis. As I mentioned in my last report, it's well known that one can use many pieces of the film samples to enhance the signal for deriving the magnetic structure. Also, one can detect the spin-split electronic structure by ARPES and compare it with that shown in this work. The authors should be positive and suggest potential experimental verifications of the magnetic order in this part.

We have edited the section V of the SI to be more forward looking and suggest potential next experiments which could significantly advance our understanding of this material. (red colored text in the SI).